# A modified fluctuation assay reveals a natural mutator phenotype that drives mutation spectrum variation within *Saccharomyces cerevisiae*

Pengyao Jiang[1], Anja R Ollodart[1,2], Vidha Sudhesh[1], Alan J Herr[3Deceased], Maitreya J Dunham[1], Kelley Harris[1,4]*

[1]Department of Genome Sciences, University of Washington, Seattle, United States; [2]Molecular and Cellular Biology Program, University of Washington, Seattle, United States; [3]Department of Laboratory Medicine and Pathology, University of Washington, Seattle, United States; [4]Department of Computational Biology, Fred Hutchinson Cancer Research Center, Seattle, United States

**Abstract** Although studies of *Saccharomyces cerevisiae* have provided many insights into mutagenesis and DNA repair, most of this work has focused on a few laboratory strains. Much less is known about the phenotypic effects of natural variation within *S. cerevisiae*'s DNA repair pathways. Here, we use natural polymorphisms to detect historical mutation spectrum differences among several wild and domesticated *S. cerevisiae* strains. To determine whether these differences are likely caused by genetic mutation rate modifiers, we use a modified fluctuation assay with a *CAN1* reporter to measure de novo mutation rates and spectra in 16 of the analyzed strains. We measure a 10-fold range of mutation rates and identify two strains with distinctive mutation spectra. These strains, known as AEQ and AAR, come from the panel's 'Mosaic beer' clade and share an enrichment for C > A mutations that is also observed in rare variation segregating throughout the genomes of several Mosaic beer and Mixed origin strains. Both AEQ and AAR are haploid derivatives of the diploid natural isolate CBS 1782, whose rare polymorphisms are enriched for C > A as well, suggesting that the underlying mutator allele is likely active in nature. We use a plasmid complementation test to show that AAR and AEQ share a mutator allele in the DNA repair gene *OGG1*, which excises 8-oxoguanine lesions that can cause C > A mutations if left unrepaired.

*For correspondence:
harriske@uw.edu

[Deceased]†Deceased

Competing interest: The authors declare that no competing interests exist.

## Introduction

Mutations are a double-edged sword. At the molecular level, they usually arise as a spontaneous consequence of DNA replication errors or damage and are the ultimate cause of genetic diseases (*Iossifov et al., 2014*; *Sebat et al., 2007*; *Antonarakis and Beckmann, 2006*; *Crow, 1997*; *Nei, 1983*). All organisms have evolved complex mechanisms for keeping mutation rates low and safeguarding their genetic information as it is passed from generation to generation (*Beckman and Loeb, 1993*; *Eisen and Hanawalt, 1999*); in multicellular organisms, these mechanisms also safeguard somatic tissues from mutations that can cause cancer and age-related decline (*Alexandrov et al., 2013*; *Loeb, 2016*; *Risques and Kennedy, 2018*). A low mutation rate is essential for long-term population survival, and the larger and more complex a genome is, the lower the mutation rate must be to prevent deleterious mutations from arising faster than natural selection can eliminate them (*Eigen, 1971*; *Drake, 1991*; *Sung et al., 2012a*; *Acosta et al., 2015*). Over long time scales, however, mutations also serve as the

raw material for evolution. Although beneficial mutations are rare occurrences, they are essential for the acquisition of novel phenotypes and adaptations (*Gompel et al., 2005*; *McGregor et al., 2007*).

A large body of theory has been written to describe how natural selection might act on the mutation rate to balance these beneficial and deleterious effects (*Sturtevant, 1937*; *Kimura, 1967*; *Leigh,, 1970*; *Johnson, 1999*; *André and Godelle, 2006*; *Sung et al., 2012a*). One prediction is that organisms living in more changeable environments might evolve higher mutation rates than organisms living in more stable environments, assuming that the environment determines whether a higher rate of beneficial mutations is likely to counterbalance a higher rate of deleterious mutations. This prediction has been borne out in laboratory evolution experiments, where mutator phenotypes sometimes emerge in populations that are forced to tolerate challenging conditions (*Good et al., 2017*; *Tenaillon et al., 2016*; *Couce et al., 2017*) and mutator strains are often observed to take over chemostat populations by producing beneficial mutations at a higher rate than competing non-mutator strains (*Chao and Cox, 1983*). Many mutator phenotypes in *E. coli* have been linked to defects in DNA repair enzymes (*Horst et al., 1999*). Mutator phenotypes also commonly occur in cancer (*Loeb, 2001*; *Prindle et al., 2010*), likely either because of relaxed selection against cellular dysfunction or because it is beneficial for cancer cells to adapt rapidly to their aggressive growth niche. However, it is less clear how much mutation rate variation exists within and between natural populations, and if such variation exists, whether it is maintained by natural selection. The drift barrier hypothesis predicts that mutator alleles will usually be deleterious because they produce more damaging mutations than beneficial ones, but that mutator alleles with relatively small effects may persist in populations because they are not deleterious enough to be efficiently eliminated (*Lynch et al., 2016*).

Although next-generation sequencing has rapidly increased our ability to measure the genetic variation that currently exists within populations, the extent of mutation rate variation is still more difficult and expensive to measure. One of the original methods for measuring mutation rates is the Luria-Delbrück fluctuation assay (*Lang and Murray, 2008*; *Gou et al., 2019*; *Luria and Delbrück, 1943*), in which a population of microorganisms is allowed to grow clonally for a controlled length of time, then challenged with a form of artificial selection that kills most cells except for those that have happened to acquire specific resistance mutations. The mutation rate can then be calculated from the number of colonies that manage to grow after this artificial selection is imposed.

Although fluctuation assays are an elegant and efficient way for measuring the mutation rates of specific reporter genes, the results are potentially sensitive to the reporter gene being used and where it is located within the genome (*Lang and Murray, 2008*; *Lang and Murray, 2011*); in addition, they are not applicable to multicellular organisms. These drawbacks have motivated the development of newer methods that take advantage of high-throughput sequencing, such as mutation accumulation (MA) assays in which a laboratory population is serially bottlenecked for many generations, eliminating most effects of natural selection and allowing mutations to be directly counted by sequencing at the end of the experiment. MA studies have been used to estimate mutation rates in a wide variety of organisms (*Lynch et al., 2008*; *Sharp et al., 2018*; *Zhu et al., 2014*; *Farlow et al., 2015*; *Wang et al., 2019*; *Keightley et al., 2009*), including a recent study of nine heterozygous *S. cerevisiae* strains (*Dutta et al., 2021*). However, higher throughput MA experiments require DNA repair genes to be knocked out to induce higher mutation rates that can be accurately measured with fewer generations of labor-intensive propagation, as in *Liu and Zhang, 2021*.

An alternative source of information about mutational processes in strains with low mutation rates is genetic variation among related individuals who share common ancestors. Polymorphic sites are easier and cheaper to discover than new mutations, because they are present at a higher density within the genome and often shared among several individuals. Mining polymorphisms for information about mutation rates can be difficult since their abundance is affected by genetic drift and natural selection (*Scally and Durbin, 2012*; *Ségurel et al., 2014*; *Zhu et al., 2017*), but despite these limitations, they have provided surprisingly strong evidence for the existence of historical changes to the mutation spectrum, meaning the tendency of mutations to occur most often in certain nucleotide contexts (*Hwang and Green, 2004*). In humans, for example, Europeans and South Asians have a significantly higher proportion of TCC> TTC mutations than other human groups (*Harris, 2015*; *Harris and Pritchard, 2017*), a pattern that is difficult to explain without a recent population-specific increase in the rate of this type of mutation. This pattern might have been caused by either a genetic mutator

or an environmental mutagen, but is not explicable by the action of selection or drift or any other process that simply modulates the retention or loss of genetic variation.

Polymorphism data has revealed that each human population and great ape species appears to have a distinctive triplet mutation spectrum, which implies that genetic and/or environmental mutators likely emerge relatively often and act within localized populations to increase mutation rates in specific sequence contexts (*Harris and Pritchard, 2017*; *Goldberg and Harris, 2021*). However, identifying these hypothetical mutators is a challenging proposition, not least because some population-specific signatures such as the human TCC> TTC enrichment appear to be relics of mutators that are no longer active. A recent study of de novo mutations in diverse human families found some evidence of mutation rate variation between human populations (*Kessler et al., 2020*), but argued that most of this variation was driven by the environment rather than genetics. Given that humans from different populations tend to be born and raised in different environments, it is extremely challenging to determine the degree to which genetics and/or the environment are responsible for variation of the rates and spectrum of de novo mutations accumulating within human populations today.

More is known about the genetic architecture of mutagenesis in model organisms, including the single-celled organism *Saccharomyces cerevisiae,* where it is tractable to disentangle genetic mutator effects from environmental ones by accumulating mutations on different genetic backgrounds in controlled laboratory environments (*Serero et al., 2014*; *Lang et al., 2013*; *Stirling et al., 2014*; *Huang et al., 2003*; *Herr et al., 2011*). Many *S. cerevisiae* mutator alleles have been discovered using genetic screens, which involve creating libraries of artificial mutants in the lab and determining which ones have high mutation rates (*Stirling et al., 2014*). Mutation rates can be elevated by up to a thousand-fold in lines where DNA proofreading and repair capabilities are artificially knocked out (*Herr et al., 2011*; *Serero et al., 2014*; *Lang et al., 2013*), and quantitative trait loci with more modest effects have been found to underlie a five-fold range of mutation rate variation among a few natural *S. cerevisiae* strains (*Gou et al., 2019*). A more complex mutator phenotype has been observed as a result of epistasis between two incompatible alleles found as natural variation in the mismatch repair genes *MLH1* and *PMS1*, although the natural isolates in which these alleles are found appear to have acquired compensatory variants that suppress this mutator phenotype (*Bui et al., 2017*; *Argueso et al., 2003*; *Raghavan et al., 2018*; *Heck et al., 2006*). An extreme case of long-term maintenance of a hypermutation lineage comes from phylogenetic analysis of bipolar budding yeasts *Hanseniaspora* which shows large amounts of gene loss, accelerated evolutionary rates and altered mutation spectra (*Steenwyk et al., 2019*).

Mild environmental stressors, such as high salt, ethanol, and heat, can also alter the mutation rate of *S. cerevisiae* (*Liu and Zhang, 2019*; *Voordeckers et al., 2020*) as well as *Arabidopsis thaliana* (*Jiang et al., 2014*; *Belfield et al., 2021*). The same environmental perturbations can cause detectable changes to both species' mutation spectra. The mutation spectrum has also been observed to depend on whether *S. cerevisiae* is replicating in a haploid or diploid state (*Sharp et al., 2018*). In addition, environmental mutagens, more complex ploidy, and genetic mutation rate modifiers could all conceivably affect the mutation spectrum of natural variation as it accumulates. However, no study to our knowledge has looked at whether any mutational signatures measured in the laboratory are capable of explaining natural mutation spectrum variation observed in polymorphism data from a model species.

Recently, comprehensive sampling efforts have produced a collection of 1011 natural isolates of *S. cerevisiae* (*Peter et al., 2018*). This is a uniquely powerful system containing abundant natural variation that accumulated within diverse environments during the recent and ancient evolution of *S. cerevisiae*, and the panel is also amenable for experimental accumulation of mutations over laboratory growth. Many genetic polymorphisms differentiate these strains, and these are relics of mutations that accumulated over many generations on divergent genetic backgrounds adapted to diverse environmental conditions, ranging from forests to beverage fermentation pipelines. Both environmental mutagens and genetic mutators may have created differences among the mutation spectra of these 1011 strains, but only genetically determined mutation spectrum differences should have the potential to be reproduced in the spectra of mutations accumulated in a controlled lab environment.

We hypothesized that yeast strains with outlying spectra of natural polymorphisms are more likely to have distinct de novo mutation spectra than strains whose polymorphisms have indistinguishable mutation spectra. The same hypothesis underlies previous inferences of de novo mutation spectrum

variation from polymorphism data (*Harris, 2015*; *Harris and Pritchard, 2017*; *Dumont, 2019*; *Goldberg and Harris, 2021*), and rare mutations found in mouse colonies (*Dumont, 2019*), but this hypothesis has been difficult to test in vertebrate species. To enable such direct testing in *S. cerevisiae*, we describe a new Luria-Delbrück-based assay that efficiently measures the spectra of de novo mutations in haploid strains using pooled amplicon sequencing. We then use this assay to identify strains with reproducibly measurable mutator phenotypes that explain the spectrum biases of these strains' polymorphisms. Some proportion of natural mutation spectrum variation might not be reproducible in the lab if it is driven by environmental mutagens, bioinformatic artifacts, or extinct genetic mutators, but our assay has the potential to identify which gradients of mutation spectrum variance are driven by extant genotypic differences.

## Results

### The mutation spectrum of natural variation in *S. cerevisiae*

To measure the mutation spectrum of genetic variation present in the 1011 *S. cerevisiae* natural isolates (*Peter et al., 2018*), we polarized single nucleotide polymorphisms using the outgroup *S. paradoxus* (*Yue et al., 2017*), then classified them into two transition types and four transversion types based on their ancestral and derived alleles. Closely related strains were excluded to avoid overrepresentation of certain groups (Materials and methods). We calculated the proportion of each mutation type among the derived alleles present in each individual strain, utilizing all derived variants present below 50 % frequency (a total of 1,213,508 SNPs). In order to minimize bias from ancestral allele misidentification, we excluded strains with extensive, pre-documented introgression from *S. paradoxus* (*Peter*

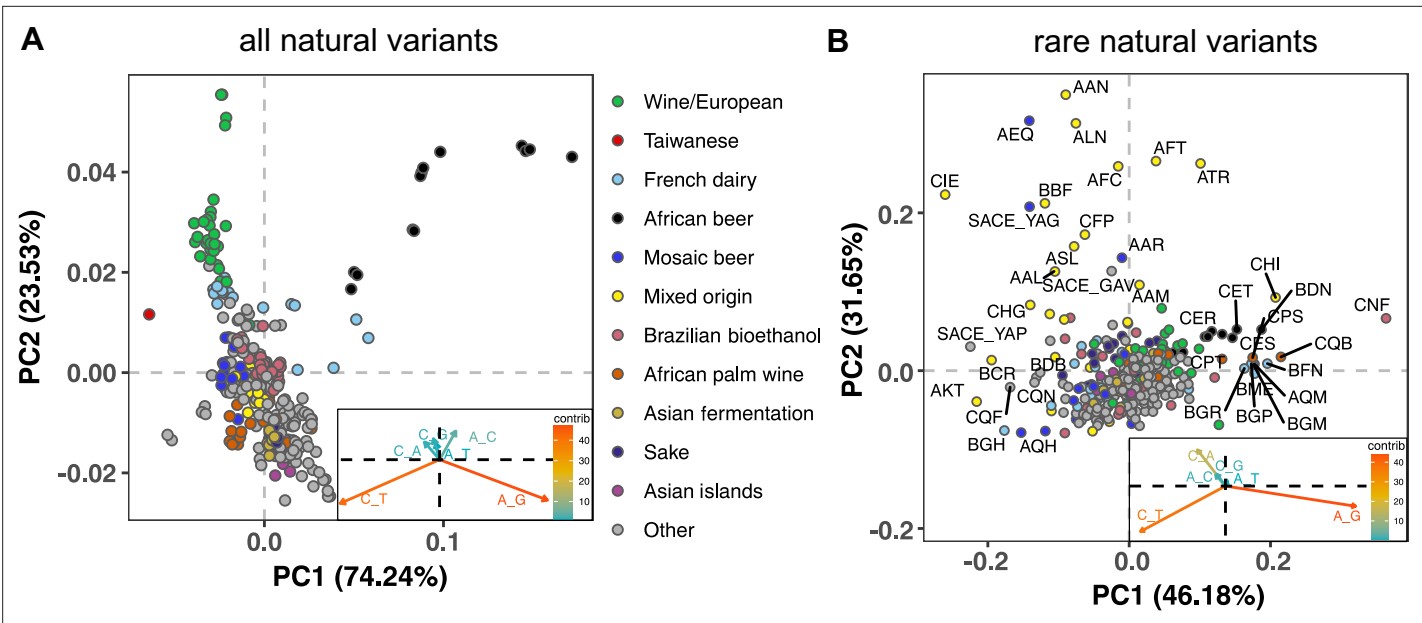

**Figure 1.** Mutation spectra of natural isolates of *S. Cerevisiae*. Principal component analysis of segregating mutation spectrum variation from a subset of the 1011 yeast strains. (**A**). Mutation spectrum PCA of all natural variants under 50 % derived allele frequency. Each strain's mutation spectrum histogram is projected as a single point, colored to indicate its population of origin (*Peter et al., 2018*). The inset summarizes the loadings of the first and second principal component vectors. (**B**). Mutation spectrum PCA of rare variants (derived allele count 2–4). Singleton variants are excluded to minimize the impact of sequencing error. Strains appearing more than 1.8 standard deviations from the origin along both PC1 and PC2 are labeled with their strain names.

The online version of this article includes the following figure supplement(s) for figure 1:

**Figure supplement 1.** Mutation spectrum PCA after subsampling to avoid overlap between lineages.

**Figure supplement 2.** PCA of synonymous variant mutation spectra.

**Figure supplement 3.** Mutation PCA using all variants stratified by triplet context.

**Figure supplement 4.** PCA of singleton mutation spectra.

**Figure supplement 5.** Mutation spectrum comparison of natural variants versus de novo mutations from a previous mutation accumulation (MA) study.

*et al., 2018*). A final set of 729,357 SNPs with ancestral alleles that passed the derived allele frequency filter were used for the following analysis (a subset of the 838,568 SNPs with callable ancestral alleles). Principal component analysis (PCA) on these individual mutation spectra reveals that strains from the same population tend to have more similar mutation spectra than more distantly related strains (*Figure 1A*, *Supplementary file 1A*). Some of this structure disappears when SNPs are subsampled to eliminate double-counting of variants that are shared among multiple strains (*Figure 1—figure supplement 1*), but several clades appear as consistent outliers in both analyses, including the African beer and European wine strains. The compact architecture of the yeast genome makes it infeasible to exclude coding regions and conserved regions, but we obtain similar PC structures when we use only synonymous protein-coding variants and when we use all polymorphisms passing quality filters (*Figure 1—figure supplement 2*).

*Figure 1A* shows that the Taiwanese and African beer populations are outliers along PC1. As seen from the principal component loadings, these two groups mainly differ from the rest in the relative proportions of the two transition types (A > G and C > T): Taiwanese strains are enriched for C > T mutations, whereas African beer strains are enriched for A > G mutations. In contrast, PC2 separates the majority of other populations, such as human-associated strains isolated from wine, dairy, and bioethanol production, along a gradient of varying transition/transversion ratio.

Although strains from the same population tend to cluster together, this trend is less pronounced in the 1,011 *S. cerevisiae* genomes than in previously reported mutation spectrum PCAs of humans, great apes, and mice (*Harris and Pritchard, 2017*; *Dumont, 2019*; *Goldberg and Harris, 2021*). That being said, one methodological difference from these previous studies is that we only partition the yeast mutation spectra into six basic types (A > C, A > G, etc.) rather than the 96 trinucleotide-based types used in analyses of vertebrate mutation spectra, a concession to the small size of the yeast genome. We found that the trinucleotide mutation spectra of yeast exhibit similar PCA structure (*Figure 1—figure supplement 3*), but that the sparsity of yeast triplet spectra appears to limit their utility.

*Figure 1B* shows a PCA of rare variant mutation spectra from the same collection of strains used in 1 A. We define rare variants as those with derived allele counts of 2, 3, and 4 (423,312 SNPs out of 729,357 fall within this allele count range) and exclude singletons to minimize the impact of sequencing error. These spectra are noisier than the spectra computed from variants up to 50 % frequency, but are potentially more likely to reflect recently active mutational processes. While rare variant spectra exhibit less clustering by population than those in *Figure 1A*, a subset of strains from several groups appear as outliers. For example, a few Mixed origin and Mosaic beer strains are outliers along a C > A mutation gradient, and African beer and French dairy strains separate out along an A > G mutation gradient. For completeness, we also examined mutation spectra of singleton variants alone (Materials and methods), which are the youngest mutations among polymorphisms (*Figure 1—figure supplement 4*). It resembles the PCA of non-singleton rare variants, except that C > A mutation variation explains a larger fraction of variation and becomes the PC1 axis.

Several previous studies have found a puzzling discrepancy between the spectra of de novo mutations and polymorphisms in *S. cerevisiae*: polymorphisms have a transition-to-transversion (ts/tv) ratio around 3:1, compared to only 1:1 for de novo mutations (*Agier and Fischer, 2012*; *Zhu et al., 2017*). Our analysis of the 1011 strain collection replicates this finding (*Figure 1—figure supplement 5*). We also replicate the prior finding that singletons and higher frequency variants have nearly identical ts/tv ratios, but that singletons inferred to be young based on their presence on long shared haplotypes have a lower ts/tv ratio somewhat closer to that of new mutations (*Figure 1—figure supplement 5*). We note that a discrepancy between de novo and segregating mutation spectra has also been observed in *A. thaliana*, though in the opposite direction, as *A. thaliana* de novo mutations are surprisingly enriched for C > T transitions (*Weng et al., 2019*).

## A scalable experimental pipeline for measuring de novo mutation rates and spectra

In order to test whether any of the mutation spectrum differences evident from natural variation in different *S. cerevisiae* strains are driven by extant genetic mechanisms that increase the rates of specific mutation types, we set out to measure several strains' de novo mutation spectra and rates experimentally. To this end, we developed an experimental pipeline using the reporter gene *CAN1*.

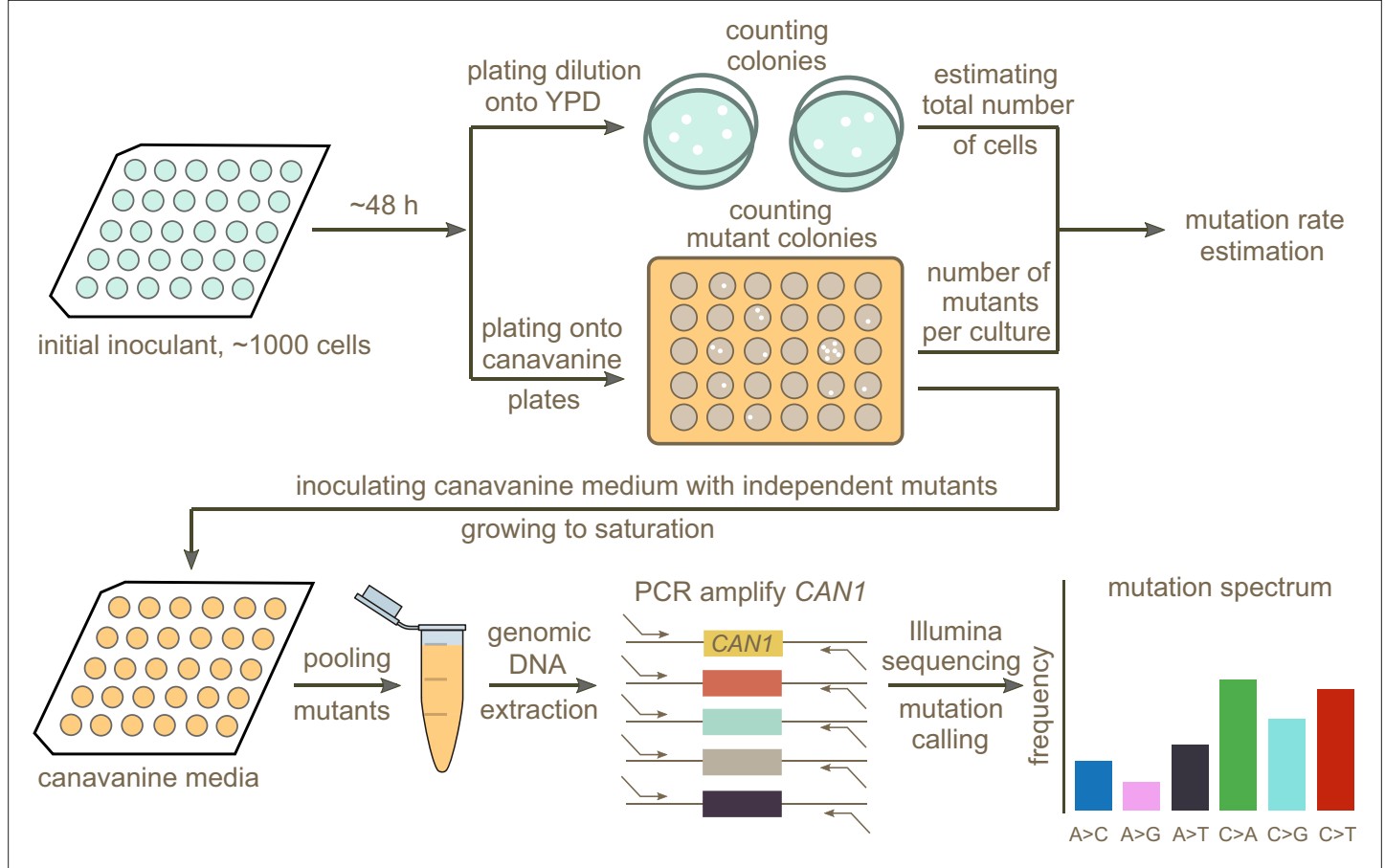

**Figure 2.** Schematic overview of the experimental pipeline. Overview of the experimental pipeline used to estimate the mutation rate and spectrum for each strain using the reporter gene *CAN1*. First, mutation rates were estimated using fluctuation assays. Independent mutants were then pooled and sequenced to estimate the mutation spectrum of each strain.

Traditional reporter gene fluctuation assays only estimate the overall rate of mutations, but we introduced an extra step that utilizes Illumina sequencing of pooled amplicons derived from *CAN1* mutants to estimate each strain's mutation spectrum as well.

The gene *CAN1* encodes a transport protein that imports arginine and arginine analogs into yeast cells from the surrounding growth media. This means that strains with a functional *CAN1* transporter are sensitive to poisoning by the arginine analog canavanine, while a single loss-of-function mutation can render such cells able to survive on canavanine media (**Whelan et al., 1979**). Poisoning a culture with canavanine is thus a very efficient method to select for cells with point mutations in *CAN1*. A limitation of this method is that it only works on genomes that contain exactly one functional copy of *CAN1*, since canavanine resistance is recessive. This means that it cannot be used to measure mutation rates in diploid or polyploid strains directly, which unfortunately include the Taiwanese strains and African beer strains that are PCA outliers in *Figure 1A*. However, 133 of the 1011 strains are haploid, leaving many strains of interest that are amenable to the assay, including several outliers in the rare variant PCA (*Figure 1B*).

A schematic overview of our experimental setup is shown in *Figure 2*. First, we estimated mutation rates using established fluctuation assay methodology (**Lang and Murray, 2008**; **Gou et al., 2019**), which involves plating multiple independent cultures from each strain being investigated. After plating, we picked a single colony from each plated culture and grew it to saturation in canavanine-containing media. We then selected mutants observed to grow in culture to similar saturation density and pooled them in equal proportions to give each mutant a roughly equal frequency in the pool. Individual pools of mutants from each strain were then subjected to PCR amplification of *CAN1* followed by Illumina sequencing. Individual mutants were called from the sequencing pools using a customized

pipeline (Materials and methods). Mutations collected from different pools of the same strain were combined to calculate the strain's mutation spectrum. We aimed to collect roughly 300 mutants per strain, enough to detect mutation spectrum differences of the magnitude estimated from polymorphism data in several of the 1011 genomes populations.

Pooling mutants across canavanine media cultures before sequencing allowed us to efficiently estimate mutation spectra at scale, yielding measurements of many individual mutations per library prep. However, pooling too many mutants during this step could have the potential to compromise the pipeline's accuracy by putting the frequency of each mutation too close to the expected frequency of Illumina sequencing errors. To test for this failure mode, we Sanger-sequenced 38 independent mutants generated using the lab strain LCTL1 (SEY6211-MATα). After pooling these 38 mutants, we performed two replicate library preps and Illumina-sequenced both using our standard procedure. Sanger sequencing identified 37 mutants with single nucleotide mutations plus one containing two adjacent mutations (*Supplementary file 1*). We expect each of these mutations, which should be present at a frequency of about 1/38 in the pooled culture, to be easily distinguishable from Illumina sequencing errors that occur at a rate of less than 1 % per base.

To identify bona fide mutations from each Illumina sequencing pool, we developed a pipeline designed to call mutations present at or above an expected frequency that is inversely proportional to the number of mutants being pooled. In order to minimize false positive mutation calls introduced by sequencing errors, we excluded low coverage regions located at the ends of the amplicons (Materials and Methods). We also identified multinucleotide mutations (MNMs) based on the co-occurrence of variation on the same reads (*Averof et al., 2000*; *Schrider et al., 2011*), separating these complex mutations from single base substitutions and small indels. When we tested this Illumina sequencing pipeline on the same mutant pool (~17,000 x coverage) that we had previously Sanger sequenced, we detected 37 of the 38 mutations identified by Sanger sequencing, missing only one mutation that occurred at the end of the amplicon located outside our pipeline's callable region. A second Illumina sequencing replicate measured only 36 of these mutations, missing one additional true mutation. Neither Illumina replicate produced any false positives, verifying that the pipeline is accurate enough to permit pooling of up to 40 *CAN1* mutants before each library prep.

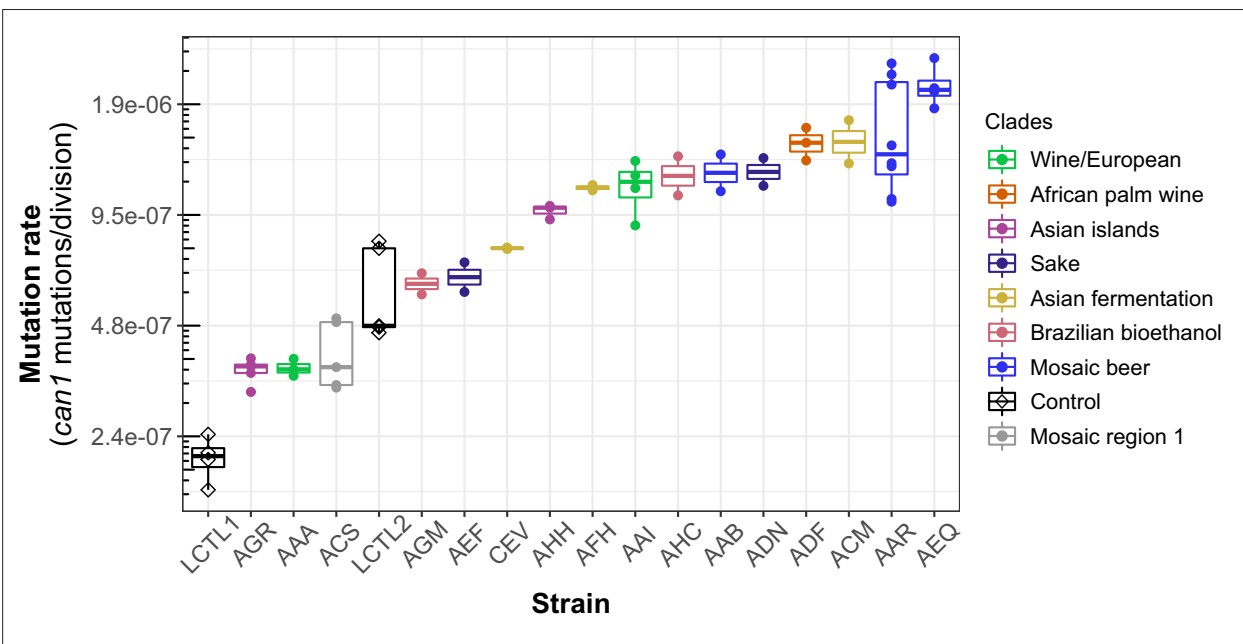

**Figure 3.** Haploid natural isolates exhibit a 10-fold range of mutation rate variation. Mutation rate variation measured among haploid natural isolates using our *CAN1* reporter gene Luria-Delbrück fluctuation assays. Strains are shown ordered by their mean mutation rates. Mutation rates for each strain were estimated using at least two replicates, each estimate represented here by a dot. A standard boxplot spans the interquartile confidence interval of possible mutation rates for each strain.

The online version of this article includes the following figure supplement(s) for figure 3:

**Figure supplement 1.** Relationship between growth rate versus measured mean mutation rate.

## Mutation rate variation among haploid natural isolates

We used our pipeline to measure mutation rates and spectra in 16 haploid strains from a wide variety of environments (*Supplementary file 1*). We selected these strains such that to the extent possible, they include two euploid strains per clade without any copy number variation at the scale of whole chromosome arms. We also selected two lab strains, LCTL1 and LCTL2, to use as controls, since their mutation rates were previously measured: the mutation rate of LCTL1 was measured using a genome-wide mutation accumulation assay by *Sharp et al., 2018*, while the mutation rate of LCTL2 (GIL 104, a derivative of W303) was measured by *Lang and Murray, 2008* using a *CAN1* fluctuation assay. Two additional strains from the 1011 collection, AAA and ACS, were selected because their mutation rates had been previously measured in another study (*Gou et al., 2019*).

We observed a 10-fold range of mutation rate variation in *CAN1* among the strains we surveyed: from $2.1 \times 10^{-7}$ to $2.1 \times 10^{-6}$ canavanine resistance mutations per gene per cell division (*Figure 3*). This range of variation is larger than the five-fold range of mutation rate variation found among six *S. cerevisiae* strains in a recent study (*Gou et al., 2019*). All estimates from different replicates of the same strain were generally consistent with each other, though three strains (ACS, LCTR2, and AAR) showed close to a twofold difference between our highest and lowest mutation rate measurements. This is within the margin of error observed in previously published fluctuation assays performed at large scale (*Gou et al., 2019*). Among the three strains (LCTL2, AAA, and ACS) with previously published mutation rate measurements, our results fall within a 1.5-fold range of those estimates, with no particular trend of upward or downward bias.

We noticed that AAR and AEQ, the two strains with the highest mutation rates, formed larger colonies during the fixed duration of the experimental growth period compared to many of the other strains tested. This suggests that AAR and AEQ may have high growth rates. This motivated us to test for correlation between the mutation rates we measured and strain-specific growth rates reported in the literature (*Peter et al., 2018*), but overall, we found no significant correlation between these attributes ($R^2 < 0.001$; p = 0.95) (*Figure 3—figure supplement 1*).

We observed that strains from the Mosaic beer, Sake, African palm wine, and Asian fermentation clades exhibited higher mutation rates than have been previously reported for any natural *S. cerevisiae* strains. The two strains with the highest mutation rates, roughly 10-fold higher than that of the control strain LCTL1, were AAR and AEQ, both from the Mosaic beer clade. While this is milder than some mutator phenotypes that have been artificially generated in the lab, to our knowledge, no comparably high mutation rate has been previously reported in a natural isolate of *S. cerevisiae*, with the exception of the spore derivatives of the incompatible *cMLH1-kPMS1* diploid natural isolate (*Raghavan et al., 2018*).

## *CAN1* sequencing reveals de novo mutation spectrum differences

We identified a total of 5571 *CAN1* mutations across all strains, including 4561 point mutations, 837 indels (*Supplementary file 1*), and 173 multinucleotide mutations (MNMs) (*Supplementary file 1*). Ninety percent of the observed indels are single base-pair indels (754 out of 837), and for simplicity we included only single base-pair indels along with point mutations when reporting each strain's mutation spectrum.

Two of the 4561 point mutations occurred at strain-specific non-reference sites. The remaining 4559 mutations consisted of repeated observations of only 727 unique mutations at 476 positions in *CAN1*. Given that each mutation was observed an average of 6.2 times, our dataset likely contains a large fraction of the mutations that are able to knock out *CAN1*'s functionality. While 23 % of the mutations are observed only once, there are several hotspots that are mutated many times in our dataset (*Figure 4—figure supplement 1*, *Figure 4—figure supplement 2*, *Supplementary file 1*). For example, genomic location 32,399 on chromosome V was hit more than 100 times across different genetic backgrounds. Eighty-five of these hits created the same G > T mutation, while 19 created another G > C mutation. Across all mutations counted with multiplicity, we observed 2676 missense mutations, 1866 nonsense mutations and only 17 synonymous mutations. These synonymous mutations made up less than 0.37 % of the total point mutations observed. Since synonymous mutations are considered a priori unlikely to cause *CAN1* to lose functionality, these variants are likely sequencing errors or hitchhikers that occurred in cells containing other inactivating mutations, or possibly rare examples of synonymous mutations disrupting translation

(*Arthur et al., 2015*) in *CAN1*. This low synonymous mutation rate further demonstrates the accuracy of our pipeline.

We were able to identify MNMs, complex mutation events that create multiple nearby substitutions or indels at once (*Stone et al., 2012*; *Harris and Nielsen, 2014*; *Averof et al., 2000*; *Schrider et al., 2011*), by looking for the presence of multiple mutations on the same Illumina read (Materials and methods). We estimate that MNMs generate 3.1 % of all point mutations, similar to the 2.6 % previously reported in a single strain background (*Lang and Murray, 2008*). Most of our strains have similar ratios of MNMs to single base-pair mutations, but there exist a few outliers (*Figure 4A*). For example, strain AAB has disproportionately many MNMs (*Figure 4A* and *Figure 4—figure supplement 3*) while AAR and AEQ have strikingly few MNMs relative to SNPs.

To our knowledge, the largest previous *CAN1* fluctuation assay in *S. cerevisiae* observed point mutations at 102 distinct positions (*Lang and Murray, 2008*). We observed mutations at 100 of these sites as well as 376 additional sites not previously known to abrogate *CAN1* function. Among the two mutations observed by Lang and Murray that are missing from our dataset, one is located near the end of the *CAN1* amplicon in a region we exclude due to insufficient sequencing coverage in most strains. The other site is the location of a mutation changing the anticodon 'CTA' to 'TTA', which is synonymous and thus not likely to have affected *CAN1* function.

One possible contributor to the observed differences in mutation rates and spectra are differences in target size: the number of distinct *CAN1* mutations that are able to help the strain survive on canavanine media. However, we were able to rule this out as a significant contributor by comparing the ratio of observed missense to nonsense mutations among strains, which was previously used by *Lang and Murray, 2008* to estimate mutational target size. We estimate that the mutational target size varies at most two-fold among strains, from 145 bp to 299 bp (*Supplementary file 1*). *Lang and Murray, 2008* previously estimated a target size of 163 bp, falling within our range. This range is too narrow to explain the 10-fold range of mutation rate variation we observe among strains, but it is large enough that we cannot confidently translate our *CAN1*-based mutation rate estimates into genome-wide estimates of the mutation rate per base pair per generation.

We performed hypergeometric tests to determine whether the mutation spectra we measured from the two control lab strains LCTL1 and LCTL2 were distinct from those measured from haploid natural isolates and from the spectrum measured from the same LCTL2 strain by *Lang and Murray, 2008* (Materials and methods). We found the spectra of point mutations we measured from the lab strains LCTL1 and LCTL2 to be statistically indistinguishable from the spectra *Lang and Murray, 2008* obtained using Sanger sequencing of canavanine-resistant mutants (p = 0.82 for LCTL1 and p = 0.087 for LCTL2). With indels included, our LCTL1 spectrum appears significantly different from that of *Lang and Murray, 2008* (p = 0.0012, Bonferroni corrected p-value: 0.042), but the LCTL2 spectra remain indistinguishable with indels included (p = 0.0054, Bonferroni corrected p-value: 0.189).

The lab strain LCTL1 appears to have a mutation spectrum that is representative of most natural isolates (*Figure 4C*, *Figure 4—figure supplement 4*, *Figure 4—figure supplement 5*). Using Bonferroni corrected *p*-values to determine significance, we found the strains AAA, ACS, AGM, AHH, AHC, AEF, ADF, ACM, AGR, AAI, and AAB to have mutation spectra that are statistically indistinguishable from that of LCTL1. We found that CEV and AFH are distinguished only by their high proportions of insertions. ADN showed significant but subtle divergence from LCTL1 in the spectrum of single nucleotide variants (*Figure 4—figure supplement 4*, *Figure 4—figure supplement 5*).

## A natural mutator phenotype with a distinctive mutation spectrum

We identified two strains with mutation spectra that diverged strikingly from the reference LCTL1 spectrum: AEQ (p < 1e-4) and AAR (p < 1e-4), the two strains that also have 10-fold higher mutation rates than LCTL1. Both strains appear highly enriched for C > A mutations compared to LCTL1 (*Figure 4B*). The strain AAR's mutation rate estimates appear somewhat bimodal (*Figure 3*), but C > A mutations are consistently enriched in replicate pools with both lower and higher estimated mutation rates. The main spectrum difference between the two mutation rate modes appears to be a small difference in the C > G mutation proportion (*Figure 4—figure supplement 6*).

In both AEQ and AAR, the proportion of C > A mutations was measured to be elevated nearly threefold above the proportion of C > A mutations in LCTL1 and similar strains (*Figure 4B*). Remarkably, this C > A enrichment appears sufficient to explain the placement of AEQ and AAR as rare

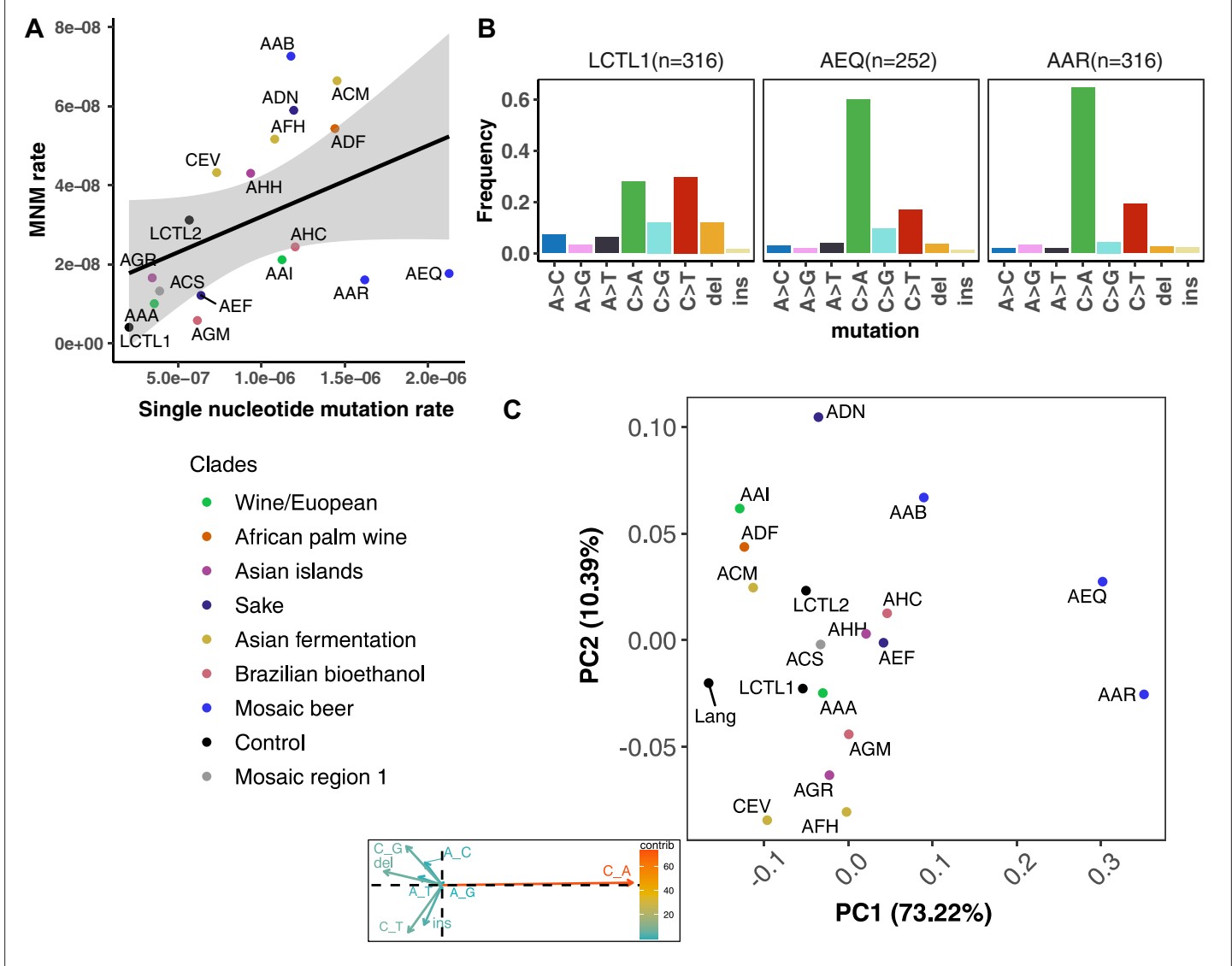

**Figure 4.** De novo mutation rates and spectra in natural isolates. (**A**). Single nucleotide mutation rates plotted against MNM rates across strains. These rates were calculated by multiplying the mean mutation rate estimated using *CAN1* by the proportion of mutations in each strain measured to be either single-nucleotide mutations or MNMs. Here, single nucleotide mutations include both single base pair substitutions and indels. (**B**). Mutation spectra in AEQ and AAR show significant enrichment of C > A mutations compared to the control lab strain LCTL1. Only single base-pair indels were used to generate these counts. (**C**). A PCA of the same strains' de novo mutation spectra compared to the mutation spectrum reported in *Lang and Murray, 2008*.

The online version of this article includes the following figure supplement(s) for figure 4:

**Figure supplement 1.** Hotspots of *CAN1* mutation across different strain backgrounds (chr V).

**Figure supplement 2.** Distribution of multiplicity of mutations observed at each mutated site in *CAN1*.

**Figure supplement 3.** Fraction of multinucleotide mutations (MNMs) in each strain.

**Figure supplement 4.** De novo mutation spectra of all strains (single base-pair substitutions and single base-pair indels).

**Figure supplement 5.** De novo mutation spectra of all strains (single base-pair substitutions only).

**Figure supplement 6.** Comparison of AAR mutation spectra from high versus low mutation rate batches.

**Figure supplement 7.** Comparison of mutation spectra from *CAN1* reporter assays versus whole genome mutation accumulation (MA).

variant mutation spectrum outliers that we previously saw in *Figure 1B*, which was computed from polymorphisms sampled genome-wide, not just within *CAN1*. Both strains have rare variant spectra that are displaced from the population norm along a principal component vector pointing in the direction of increased C > A enrichment. These strains' high mutation rate, C > A-heavy de novo mutation spectrum, and concordant C > A-heavy rare variant spectrum all point to the conclusion that these Mosaic beer strains display a naturally occurring genetically encoded mutator phenotype.

To assess whether the C > A enrichment phenotype observed in AAR and AEQ is likely shared with any of the 1,011 strains for which we lack mutation spectrum measurements, we ranked all 1011 strains (excluding close relatives) according to the C > A enrichment of their rare variants. We found that SACE_YAG and BRM, the two strains closest to AAR and AEQ in the global neighbor-joining phylogeny, clustered with AAR and AEQ in having high C > A fractions within the top 5 % in the dataset. The rest of the top-ranking 5 % of the strains exhibit some phylogenetic clustering, but no others fall within the Mosaic beer clade (*Figure 5A*). Instead, they are somewhat dispersed across two large, diverse clades known as 'Mosaic region 3' and the 'Mixed origin' clade. We also used a bootstrapping method to find strains with enriched C > A fractions, using an empirical p-value threshold of 0.05. Many of the same strains are outliers in both tests, including the four Mosaic beer strains (*Figure 5—figure supplement 1*). The phylogenetic clustering of C > A rare variant enrichment suggests that multiple clades may be genetically predisposed toward accumulating relatively higher rates of this mutation type.

Focusing on the small clade containing AAR, AEQ, BRM, and SACE_YAG with all members showing C > A enrichment (*Figure 5B*), we found that all but BRM are haploid derivatives of the diploid *Saccharomyces cerevisiae var diastaticus* strain CBS 1782, which was isolated in 1952 from super-attenuated beer (*Andrews and Gilliland, 1952*). AEQ and AAR differ at roughly 14,000 variant sites (the median pairwise genetic distance in the 1011 strains is 64,000) and SACE_YAG differs from AEQ and AAR at about 11,000 sites each. These differences among haploid derivatives reflect the high level of heterozygosity in the parental diploid strain. The fourth strain with an elevated C > A mutation fraction, BRM, is derived from an independent source: it was isolated in 1988 from a cassava flour factory in Brazil (*Laluce et al., 1988*). Despite its distinct origin, BRM differs at only 14,000–17,000 sites from each of the aforementioned three strains.

In AAR, AEQ, BRM, and SACE_YAG, rarer variants are notably C > A-enriched, but higher frequency variants exhibit weaker C > A enrichment that declines with increasing allele frequency such that variants of allele count greater than eight have C > A fractions more typical of other strains. We verified that the same pattern holds for variants present in the whole genome of the ancestral diploid isolate CBS 1782, which we sequenced after obtaining a sample from the NCYC database (see Materials and Methods). This suggests that AAR and AEQ indeed inherited their mutator phenotype from their wild ancestors and that this phenotype is at least partially penetrant in the diploid state. In contrast, the four closest outgroups to AAR, AEQ, BRM, and SACE_YAG in the 1011 yeast genomes phylogeny exhibit a consistently lower C > A fraction that does not vary with allele frequency (*Figure 5B* and *Figure 5—figure supplement 2*). The concordant enrichment of C > A mutations in rare polymorphisms and de novo mutations from the same strains suggests that this C > A enrichment is genetically determined and is not specific to the *CAN1* locus but has affected the entire genome during the recent history of this clade.

## A scan for candidate mutator alleles

To explore genetic variation that could underlie the C > A enrichment phenotype observed in AEQ and AAR, we scanned for nonsense and missense mutations in a list of 158 candidate genes known to play roles in DNA replication and repair, including genes that were previously identified to harbor mutator alleles through genetic screens (*Supplementary file 1*; *Boiteux and Jinks-Robertson, 2013*; *Stirling et al., 2014*). No candidate premature stop codons were found to be both present in AAR and AEQ and rare (MAF <0.05) in the 1011 yeast genomes overall. However, we identified 40 sites with at least one rare non-synonymous allele (MAF <0.05) shared by AEQ and AAR and absent from the other haploid strains that we experimentally found to have normal mutation spectra (*Supplementary file 1*).

One of these missense variants was observed within *OGG1*, a glycosylase involved in the oxidative stress response that excises the guanine lesion 8-oxo-G. *ogg1* null mutants are known to have high C > A mutation rates (*Shockley et al., 2013*) and mutation rates that are 10-fold elevated above normal

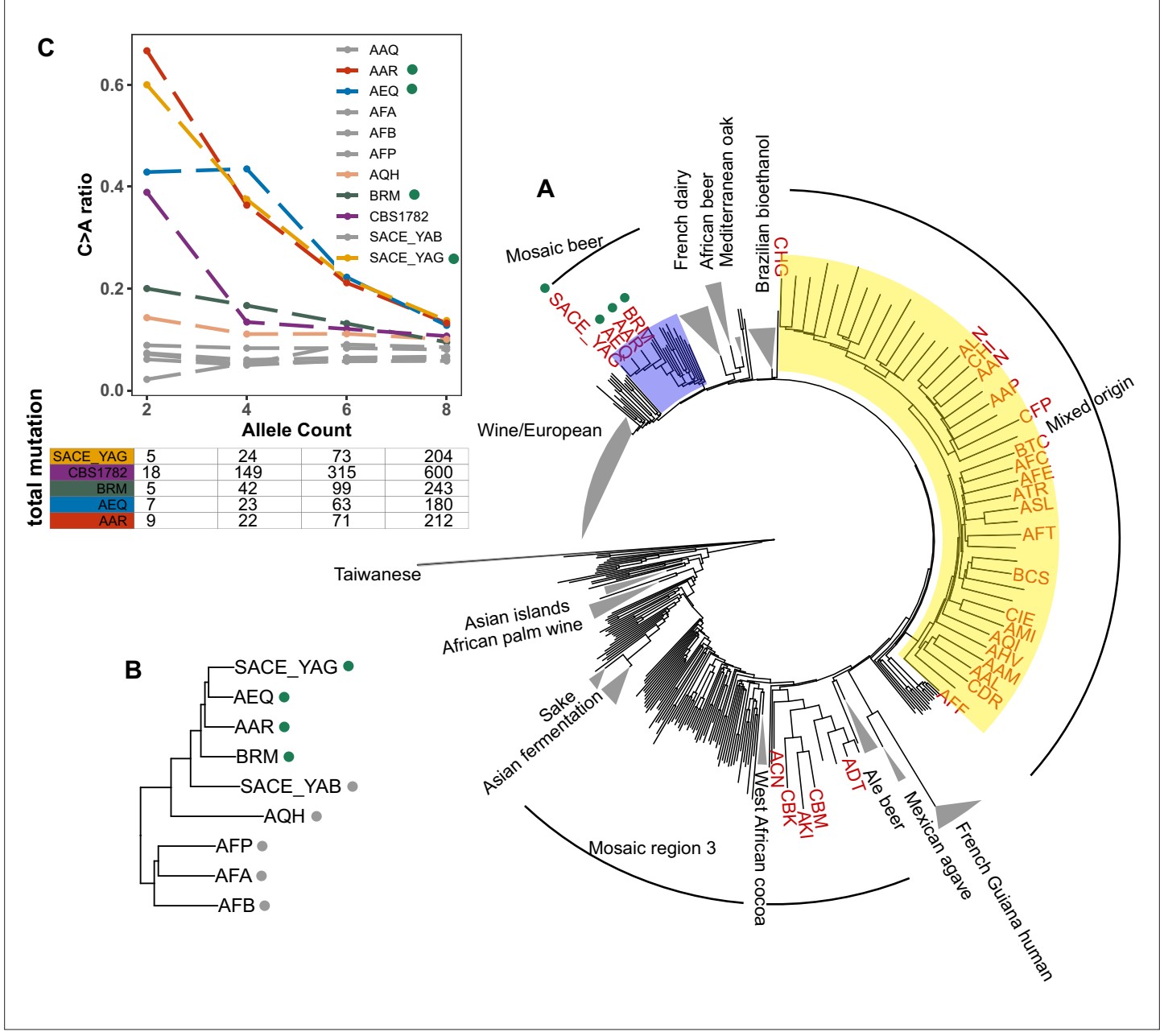

**Figure 5.** Enrichment of C > A mutations in rare natural variants. (**A**) Phylogeny of the 1011 collection with strains in the top 5 % of C > A fraction shown in red. (**B**) Phylogeny of AEQ, AAR, and closely related Mosaic beer strains (**C**) Top panel: The C > A ratio as a function of minor allele count in Mosaic beer strains that are closely related to AEQ and AAR, including AEQ's and AAR's the diploid common ancestor CBS 1782. C > A ratios in polymorphisms are calculated across allele count (AC) bins with cutoffs of 2, 4, 6, and 8. When computing allele counts, closely-related strains are excluded, and each strain is represented as a diploid in genotype. Bottom panel: total number of variants in each AC bin.

The online version of this article includes the following figure supplement(s) for figure 5:

**Figure supplement 1.** Enrichment of C > A in rare polymorphisms in the 1,011 collection (measured using empirical bootstrapping).

**Figure supplement 2.** Enrichment of C > A mutations in rare natural variants from Mosaic beer strains that are closely related to AEQ and AAR from the 1,011 collection.

**Figure supplement 3.** Mutation spectra of rare natural polymorphisms in AQH stratified by minor allele count.

(**Ni et al., 1999**). The *OGG1* missense variant present in AAR and AEQ is homozygous in the parental diploid CBS 1782; the same variant is also shared by two other Mosaic beer strains, AQH and AAQ, which are not enriched for C > A rare variants. A close examination of rare variant mutation spectra

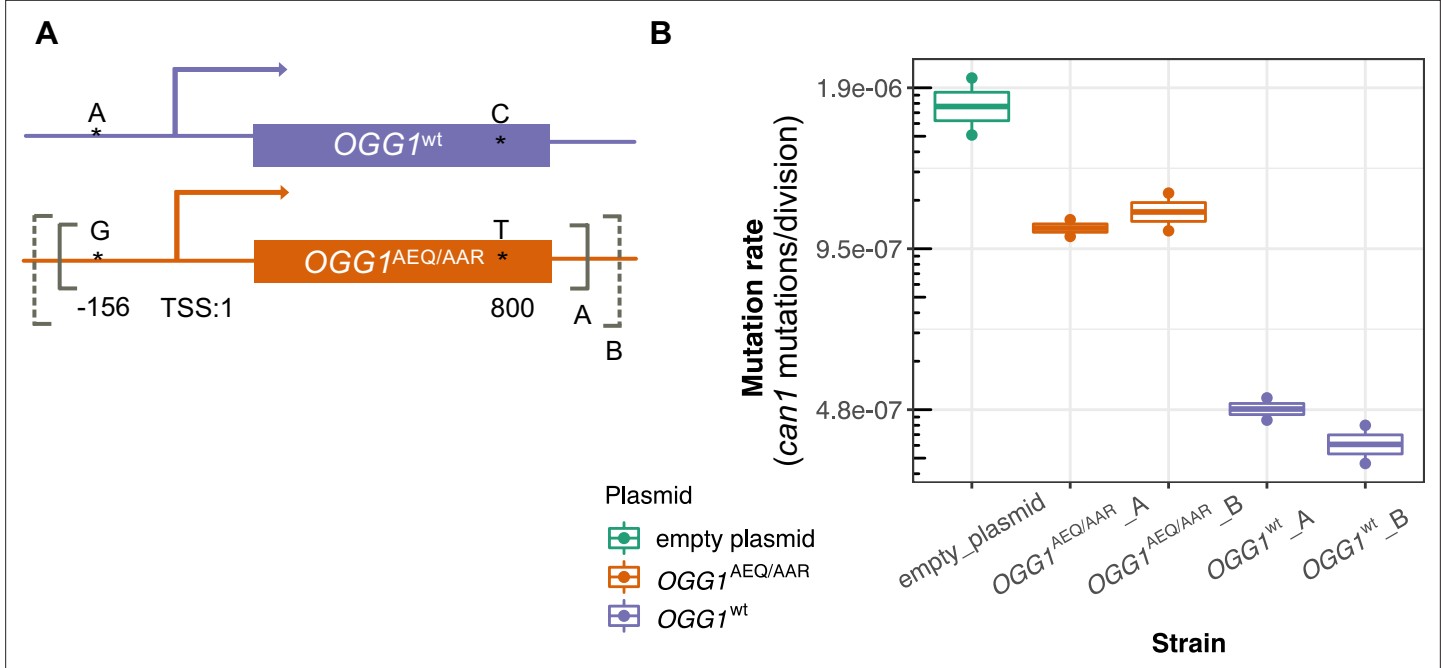

**Figure 6.** Mutation rates of an *ogg1* deletion strain complemented by *OGG1*^AEQ/AAR versus *OGG1*^wt. (**A**) Locations of the genetic variants differentiating the *OGG1*^wt and *OGG1*^AEQ/AAR alleles. Solid and dashed lines denote the boundaries of *OGG1_A* and *OGG1_B* (not to scale). *OGG1_A* contains a 1.7 kb PCR fragment including *OGG1*, while OGG1_B contains a 2.2 kb PCR fragment. (**B**) Mutation rate estimates of *ogg1* deletion strains transformed by plasmids containing different OGG1 alleles.

revealed that AQH may have a C > A enrichment phenotype that is masked by a high rate of C > T mutations (*Figure 5—figure supplement 3*), but no evidence of C > A enrichment was found in AAQ.

## Experimental evidence that *OGG1*^AEQ/AAR is a mutator allele

We sought to experimentally test whether the *OGG1* allele present in AEQ and AAR, hereafter called *OGG1*^AEQ/AAR, contributes to the AEQ/AAR mutator phenotype. We accomplished this by expressing the *OGG1*^AEQ/AAR allele in a lab strain with its endogenous *ogg1* allele deleted. If *OGG1*^AEQ/AAR contributes to the mutator phenotype of AEQ and AAR, then this gene variant should be less effective at rescuing the *ogg1* deletion phenotype than the wild-type *OGG1* allele.

To test this hypothesis, we constructed two CEN plasmids. One plasmid was constructed to express *OGG1*^AEQ/AAR; the other was constructed to express *OGG1*^wt, with the same sequence as the *OGG1* allele present in the S288C reference. Through Sanger sequencing, we found that *OGG1*^AEQ/AAR in fact contains two single nucleotide differences from the reference sequence (*Figure 6A*). One of these variants (an A > G substitution) lies 156 bp upstream of *OGG1*, presumably in the gene's promoter region, which we previously missed when examining the coding sequence. The other variant is a nonsynonymous change from C to T at position 800, which changes alanine to leucine. While the nonsynonymous mutation in *OGG1* is present in only seven strains from the Mosaic beer clade (including AEA which is a close relative of BRM) from the 1011 collection, the promoter mutation is present in 57 strains. Since the promoter mutation is more widespread and not linked to the nonsynonymous mutation, we hypothesize that it is less likely to cause the mutator phenotype. To avoid truncating any important regulatory sequences, we used PCR to amplify two fragments with different lengths of flanking sequence from each endogenous *OGG1* locus: one shorter fragment called *OGG1-A*, (1.7 kb) and one longer fragment called *OGG1-B* (2.2 kb), creating a total of four plasmids. We then used these *OGG1* plasmids, as well as an empty plasmid, to transform haploid *S. cerevisiae* with an *ogg1* deletion derived from the heterozygous diploid deletion collection (SGA) (*Boone, 2007*).

We used fluctuation assays to estimate the mutation rates of *ogg1* deletion strains complemented with these different *OGG1* plasmids. We found that the mutation rate of the *ogg1* null yeast with an empty control plasmid was elevated fourfold compared to the mutation rate of the strain transformed with *OGG1*^wt. This difference is smaller than the 10-fold mutation rate difference previously measured

between an *ogg1* null mutant and a strain with an intact endogenous *OGG1* allele (**Ni et al., 1999**), possibly reflecting a difference in the efficiency of endogenous *OGG1* expression versus expression of the same allele from a plasmid. We found that the strain transformed with *OGG1^AEQ/AAR^* had a 2.4-fold higher mutation rate than the strain transformed with *OGG1^wt^*, suggesting that the AEQ/AAR allele is less effective at complementing the DNA repair deficiency of the *ogg1* deletion strain. This supports our hypothesis that the *OGG1^AEQ/AAR^* allele is at least in part responsible for the high mutation rate of the AEQ and AAR strains. We measured similar mutation rates in strains transformed by the shorter and longer PCR fragments containing each *OGG1* allele, suggesting that the additional regulatory DNA present in the longer sequence did not substantially impact *OGG1* expression.

## Discussion

In this study, we compare the mutation spectra of natural variants and de novo mutations within diverse strains of *S. cerevisiae*. We find both population genetic and experimental evidence pointing to a recent increase in the rate of C > A mutations within one clade as a result of a naturally occurring genetically encoded mutator phenotype.

Our highly accurate pipeline uses a classical fluctuation assay to quantify de novo mutation rate in the reporter gene *CAN1,* then utilizes sequencing of pooled mutants to measure mutation spectra. Sanger validations confirmed that this approach has an undetectably low false positive rate and a very low false negative rate. The reliability of our pipeline is further supported by concordance between our mutation spectrum measurements and previous measurements made from the same lab strains (**Lang and Murray, 2008**). Using this method, we identified two extreme mutation spectrum outliers—the Mosaic beer strains AEQ and AAR—as well as several strains with more subtle mutation spectrum phenotypes that are good candidates for future investigation.

Although the mutation spectrum variation from natural polymorphisms we report here is reminiscent of observed mutation spectrum variation among humans, great apes, and laboratory mouse strains, we note that *S. cerevisiae* mutation spectra separate less by population compared to vertebrate mutation spectra. While it is possible to infer a human genome's continental group of origin using its mutation spectrum alone, the same is not generally true in *S. cerevisiae*. Several factors, which are not mutually exclusive, might underlie this difference. One is the presence of complex population structure and pervasive gene flow between *S. cerevisiae* clades (**Schacherer et al., 2009**; **Peter et al., 2018**; **Liti et al., 2009**). Another factor is the small size of the *S. cerevisiae* genome; each strain has two orders of magnitude fewer derived alleles than most vertebrate genomes have. This limits our ability to detect how flanking base pairs affect the mutation rate of each site in the genome and constrains us to study a lower dimensional mutation spectrum than the context-dependent ones studied in species with larger genomes.

While gene flow and genome size might be responsible for the relatively modest magnitude of mutation spectrum divergence between most strains of *S. cerevisiae*, it is also possible that DNA replication and repair are intrinsically more uniform in *S. cerevisiae* than in vertebrates, perhaps because of the greater efficiency of selection against weakly deleterious mutator alleles in a unicellular organism that exists at large effective population sizes and often reproduces asexually. Asexual reproduction should theoretically increase the efficiency of selection against mutator alleles because deleterious variants created by the mutator cannot recombine onto other genetic backgrounds; on the other hand, it can also limit the efficiency of selection against individual deleterious mutations by permanently tethering them to particular genetic backgrounds. Further measurements of mutation spectrum variation within other species will be needed to determine whether the stability observed here is indeed characteristic of unicellular eukaryotes. If mutation spectra tend to be stable within species that have low mutation rates and strong selection against mutation rate modifiers, we might expect to see even less mutation spectrum variation among populations of ciliates like *Paramecium* and *Tetrahymena*, whose mutation rates are substantially lower than that of *S. cerevisiae* (**Sung et al., 2012b**).

Many questions about mutation spectrum variation within *S. cerevisiae* and other species remain unresolved and present important avenues for future work. For example, it is unclear whether rare polymorphisms in the Mixed origin and Mosaic region three clades are enriched for C > A mutations due to the same genetic mechanisms active in AEQ and AAR. Other genes might underlie the mutation spectrum differences observed among other strains, though our analyses suggest that some mutation spectrum gradients that dominate the common variation PCA are unlikely to be explained

by extant mutators. One such gradient is the A > G enrichment in the African beer yeast clade, which is less pronounced in our rare variant PCA (*Figure 1B*) compared to our PCA extracted from variation of all frequencies (*Figure 1A*). This is somewhat reminiscent of the frequency distribution of the TCC> TTC mutation 'pulse' that distinguishes Europeans and South Asians from other human populations, and may suggest that the African beer A > G enrichment was caused by an extinct mutator allele or a mutagen found in a past environment.

Natural selection might contribute to the mutation spectrum variation within *S. cerevisiae* if certain mutation types are more often beneficial than others and if such asymmetries vary between populations. One example of how natural selection can affect mutation spectrum variation is that transitions are more often synonymous than transversions are (*Freeland and Hurst, 1998*), leading to more frequent selection against transversions in genic regions. However, we note that most of the gradient structure observed in our analyses can be reproduced with synonymous mutations alone, meaning that selection is unlikely to explain much of the natural yeast mutation spectrum variation we observe.

The C > A mutations enriched in AEQ, AAR, and their relatives might be a signature of oxidative stress damage; such mutations are a known signature of failure to repair 8-oxoguanine lesions, which is consistent with a causal role for these strains' missense substitution in the oxidative stress response gene *OGG1* as verified through our *OGG1* plasmid assay. A recent study knocked out *OGG1* in human cells and found that this elevated the rate of a C > A-dominated mutational signature known in the Cosmic catalog as SBS18 (*Zou, 2020*). SBS18 was initially identified in tumors from individuals with pathogenic variation in *Mutyh* (*Viel et al., 2017*), a direct interacting partner of *OGG1* in the 8-oxoguanine repair pathway that has also been implicated in a C > A-dominated germline mutational signature in mice (*Sasani et al., 2021*).

Although our *OGG1* plasmid assay provides compelling evidence that the 8-oxoguanine repair pathway plays a role in the AAR/AEQ mutator phenotype, we note that C > A mutations do not comprise all of the excess mutations measured in AEQ and AAR. Compared to LCTL1, the ratio of C > A mutations to C > T mutations is elevated roughly threefold in AEQ and AAR, which is less than the 10-fold overall elevation of the mutation rate in these strains. These mutation data imply that all mutation types have higher rates in AEQ and AAR compared to other *S. cerevisiae* strains, not just C > A. Further work will be required to verify whether the C > A enrichment and broad-spectrum mutation rate increase are driven by the same biochemical mechanism.

Although our estimates of de novo mutation spectra are all based on the *CAN1* locus, we note that *CAN1* mutation spectra are generally similar to the genome-wide spectra obtained from MA experiments (*Zhu et al., 2014*; *Sharp et al., 2018*), except that *CAN1* spectra have more mutations at C/G base pairs and relatively fewer mutations at A/T base pairs (*Figure 4—figure supplement 7*). The 2676 and 1866 missense and nonsense mutations in our dataset contain numerous instances of all six mutation types that comprise our summary mutation spectrum, suggesting that *CAN1* contains many opportunities to ascertain the full spectrum observed genomewide. We noticed that the *CAN1* spectra we measured from the LCTL2 strain showed a slightly higher proportion of C > A mutations than that measured from the same strain by *Lang and Murray, 2008* also using a *CAN1* reporter. This could be due to the fact that Lang et al. used ten times the concentration of canavanine that we used, which might mean that some mutants could have grown after plating on our canavanine media (*Shor et al., 2013*). The fact that we do not see elevated C > A mutations in LCTL1 suggests that this could be a LCTL2 strain-specific behavior.

In summary, the results presented in this paper provide some of the most direct evidence to date that eukaryotic mutation spectra are variable within species (*Harris, 2015*; *Harris and Pritchard, 2017*; *Dumont, 2019*; *Goldberg and Harris, 2021*). It has been proposed that the best explanation for such mutation spectrum heterogeneity is the frequent emergence of nearly neutral mutator alleles that turn over rapidly as a consequence of weak purifying selection on the mutation rate. Our de novo mutation spectrum measurements provide experimental verification of this claim, showing that at least one mutational signature whose activity varies among natural yeast strains is likely caused by an extant mutator allele.

Although our results show that the mutation spectrum bias shared by certain Mosaic beer yeast is genetically encoded, it is worth noting that this C > A gradient is not the principal axis of mutation spectrum variation in the 1011 yeast genomes that we computed from all variants (*Figure 1A*). It remains to be seen how many other mutator or antimutator alleles might exist within this strain

collection and to what extent they can explain the mutation spectrum variation observed among strains from different environments. A broader question still is whether the forces that created *S. cerevisiae*'s mutation spectrum variation are similar to the forces that shaped the distinctive mutation spectra of different human populations and great ape species. If we can identify the genes that underlie natural yeast mutator phenotypes such as the one described in this study, it will likely be more straightforward to test these genes for mutator activity in humans and other species than to discover mutator alleles via any kind of agnostic genome scan.

## Materials and methods

### Variant filtering and mutation PCA analysis

We filtered the original variants from the 1011 *S. cerevisiae* collection (*Peter et al., 2018*) by including biallelic SNPs with less than 20 % missing genotypes. We restricted to regions in the genomes where reads can be uniquely mapped ('mregions_100_annot_2011.bed' from *Jubin et al., 2014*) and excluded repeat-masked regions. Closely related strains (pairwise genetic distance less than 8000) are excluded from the 1011 dataset. Singletons were excluded when counting individual mutations in *Figure 1* to minimize the impact of sequencing errors. Strains with extensive introgression from *S. paradoxus* (clades 2, 9, and 10 from *Peter et al., 2018*) were excluded in order to minimize bias from errors in the inference of ancestral and derived alleles. Ancestral states of mutations were inferred using five *S. paradoxus* sequences (*Yue et al., 2017*), aligned to the *S. cerevisiae* reference genome R64-1-1 using lastz v1.04.00 (*Harris, 2007*). Only sites that are fixed in four out of five strains were inferred to be the ancestral alleles, and other sites were ignored. When computing the mutation spectra of strains from variants for *Figure 1A*, each individual strain was assumed to be diploid, with homozygous derived alleles counted with twice the weight as heterozygous derived alleles. When counting rare variants, homozygous derived alleles were given the same weights as heterozygous derived alleles.

To further minimize confounding of the mutation spectrum by ancestral allele misidentification, only variants with derived allele frequency less than 0.5 were used. Variants that passed all filtering criteria were used to compute a normalized mutation spectrum histogram for each individual strain. When performing PCAs, no more than 30 strains from each population were randomly sampled to minimize bias from uneven sampling. The same strains were used to generate PCA plots in *Figure 1—figure supplements 1–4* (*Supplementary file 1*), except that strains with fewer than eight singletons or rare variants were further excluded when generating *Figure 1—figure supplement 4* and *Figure 1B*. Our definition of singletons varied as a function of ploidy (*Figure 1—figure supplements 4–5*): In haploids and homozygous diploids (as defined in *Peter et al., 2018*), a singleton will be fixed in the strain where it occurs (represented as homozygous), but in other types of strains, a singleton is required to be heterozygous. In all cases, a singleton is a variant present in only a single strain.

### Fluctuation assays and sequencing

We performed fluctuation assays according to an established protocol (*Lang, 2018*) with the following modifications: 4µl of overnight inoculant was diluted in 40 ml SC-Arginine +2 % Glucose media. 50µl of the diluted cultures were distributed in 96-well round-bottom plates (Costar 3788) for each strain. Plates were sealed with Breathe-Easy sealing membrane (Sigma Z380059). SC-Arginine-Serine+ Canavanine (60 mg/liter L-canavanine) Omni plates (Nunc OmniTray 242811) were used and dried for 2–4 days in a 30°C incubator before using. Depending on the strains, 50µl of culture were diluted one- to fourfold when plating on the Omni plates, either to reduce the background or to avoid growth of too many mutant colonies. After plating, the plates were dried and then incubated at 30°C for 48 hr. Independent mutants from separate cultures were inoculated into 200µl SC-Arginine-Serine + 60 mg/Liter Canavanine +2 % Glucose media, and then grown to saturation over ~43 hr at 30°C with shaking. Optical densities (ODs) were measured after incubation, and only mutants that reached similar saturation ODs were pooled (150µl each) to achieve equal proportions. Genomic DNA from each pool was extracted using the Hoffman Winston protocol (*Hoffman and Winston, 1987*). *CAN1* was then PCR amplified using published primers (*Lang and Murray, 2008*) with 15 cycles. Two independent 25µl PCR reactions were then pooled and cleaned up with a Zymo Clean & Concentrator Kit (D4004). Sequencing libraries were prepared using the Nextera XT DNA Library Preparation Kit with

customized indices. Sequencing runs with 75 or 150 bp paired-end reads were performed using an Illumina NextSeq 550 sequencer (BioProject: PRJNA691686).

## Calculation of mutation rates

The rSalvador package (*Zheng, 2017*) was used to estimate the number of mutation events ($m$) in each fluctuation assay using maximum likelihood under the Lea-Coulson model (*Lea and Coulson, 1949*; *Luria and Delbrück, 1943*; *Ma et al., 1992*). The total number of cells ($Nt$) was measured by counting colonies seeded with dilutions of cells on YPD plates with dilutions ranging from 1:10,000 to 1:40,000. The rate of loss-of-function mutations per *CAN1* gene per cell division was estimated to be $m/Nt$.

## Mutation calling

Sequencing reads were first mapped using bowtie v2.2.3 (*Langmead and Salzberg, 2012*) to the Scer3 S288C reference *CAN1* PCR fragment sequence using primers designed by *Lang and Murray, 2008*. Mutation coordinates were therefore called relative to the start of the *CAN1* amplicon. Adapters were trimmed using the program trim_galore v0.6.6 (*Krueger, 2021*) and paired-end reads were merged using pear v0.9.11 (*Zhang et al., 2014*). The command `fastq_quality_filter -q 20 p 94` was used to remove low quality reads before running bowtie. A MAPQ cutoff of 40 was used for SNPs and a cutoff of 20 was used for indels. Pysamstats v1.1.2 (*Alistair, 2021*) was used to compute the frequencies of all possible alleles at each base pair. Sites with read depth less than 200 or with less than 40 % coverage of the amplicon were excluded. After the first round of mapping, sites that were fixed in each strain were called and compared to the SNPs in the 1011 collection to confirm strain identity. We then performed a second round of read mapping using the same pipeline except that each strain's reads were mapped to a strain-specific *CAN1* reference sequence.

For each sequencing pool, we let $N$ be the number of mutants that were pooled prior to sequencing. Non-reference alleles with frequencies between $0.65 \times 1/N$ and 0.95 were included as evidence of mutations, discarding alleles below this frequency range as likely to be sequencing errors and alleles above this frequency range as likely to be strain-specific SNPs. Adjacent indels were merged if their frequencies differed by less than 10%. MNMs were identified in each pool by first flagging pairs of mutations occurring at similar frequencies (plus or minus 9%) within 10 bp of one another and then verifying the coexistence of the two mutations on at least 70 % of the paired-end reads where at least one of the two mutations appears. Complex MNMs containing three or more variants were identified by merging MNMs that share an SNP in common. To obtain single nucleotide mutation counts and indel counts, mutations that are part of MNMs were first excluded from each pool. The coordinates of each mutation were converted back from *CAN1*-specific coordinates to genomic positions. Point mutations were further annotated using VEP (*McLaren et al., 2016*) to further categorize into missense, nonsense, or synonymous mutation types.

Allele frequencies were used to estimate the multiplicity of each mutant as follows: First, the mean and standard deviation of all mutant allele frequencies were calculated from each pool. Each allele frequency more than two standard deviations above the mean was then translated into a mutation count by dividing it by the mean allele frequency and then rounding to the nearest integer. Mutations with frequencies less than two standard deviations above the mean are assumed to be mutations with count 1.

## Statistically quantifying mutation spectrum differentiation

To compare the mutation spectra between strains, mutations were first classified as one of the six general classes of base-substitutions (A > C, A > G, A > T, C > A, C > G, C > T) or as single base-pair insertions or deletions. We then compared the mutation spectra of the two control strains LCTL1 and LCTL2 to all other haploid isolates as well as one spectrum published by *Lang and Murray, 2008* (a total of 35 tests) using a pairwise hypergeometric test (*Adams and Skopek, 1987*), a custom python script (*Tracy et al., 2020*). In the first round of this test, the paired mutation counts were arranged in a 2 × 8 contingency table. To test the null hypothesis that the two mutation spectra are the same, the hypergeometric probability of the observed table was calculated and compared to the hypergeometric probabilities of 10,000 random tables with the same row and column totals. The number of random tables with a higher hypergeometric probability than the observed provides an estimate

of the p-value. We used the conservative Bonferroni correction to compute the significance cutoff (0.05/35 = 0.001429). A second set of Bonferroni-corrected p-values was calculated after excluding indels to form a 2 × 6 contingency table. These p-values were used to determine how many of the significant mutation spectrum differences were driven by the indel category (*Figure 4—figure supplement 4*, *Figure 4—figure supplement 5*).

### Whole-genome sequencing and variant calling of CBS 1782

A sample of CBS 1782 was obtained from the National Collection of Yeast Cultures (NCYC 361) January of 2021. Genomic library was prepared from the Illumina Nextera DNA Flex Library Prep kit (Illumina 20018704). Sequencing runs with 150 bp paired-end reads were performed using an Illumina NextSeq 550 sequencer to roughly 250 x coverage. Sequencing reads were mapped using BWA mem v 0.7.17 (*Li and Durbin, 2009*). Only reads that are uniquely mapped with MQ >20 were used. GATK4 HaplotypeCaller was used for variant calling (*Poplin, 2018*). Variant sites (±2 bp) that overlap with deletion were filtered out. The variants for CBS 1782 were then merged with the raw vcf file for the 1011 collection excluding closely-related strains, and underwent the same filtering as for the PCA analysis.

### Constructing *OGG1* plasmids

Haploid yeast with both *ogg1* and *leu2* deleted was generated from spores dissected from the heterozygous diploid strain #20,510 of the SGA collection (*Boone, 2007*). CEN plasmids were generated using pRS415 with *LEU2* as the selection marker (*Chee and Haase, 2012*). *OGG1-A* and *OGG1-B* were amplified by PCR from strain AAR for *OGG1^{AEQ/AAR}* or from LCTL1 for *OGG1^{wt}*. Primers for *OGG1-A:* CGATAGTTTGGCGTGCGATA, CGCCTTGGTGACCGTTTT. Primers for *OGG1-B:* GGTTCTTCCCAAT-CATCCGA,AGGGCTTATTGACGACGACA. pRS415 were linearized by HindIII and BamHI, followed by Gibson assembly with the *OGG1* alleles.

## Acknowledgements

We dedicate this paper to the memory of Dr. Alan J Herr, a sorely missed colleague, mentor, friend and inspiring role model. We thank all members of the Harris and Dunham labs for helpful comments and discussions. We thank Nathaniel Sharp and Greg Lang for sharing strains. We also thank Joseph Schacherer for sharing the 1,011 strain collection with the Dunham lab. PJ was supported by a Burroughs Wellcome Fund Career Award at the Scientific Interface awarded to KH. KH acknowledges additional support from a Searle Scholarship, a Sloan Research Fellowship, a Pew Biomedical Scholarship, and National Institute of General Medical Sciences Grant 1R35GM133428-01. AJH was supported by the National Institute for General Medical Sciences (NIH/NIGMS R01GM118854). ARO was supported by the National Human Genome Research Institute of the NIH under award T32 HG00035. The research of MJD was supported by NIH/NIGMS award P41 GM103533 and a Faculty Scholar grant from the Howard Hughes Medical Institute. The content is solely the responsibility of the authors and does not necessarily represent the official views of the Burroughs Wellcome Fund, the Kinship Foundation, the Sloan Foundation, the Pew Charitable Trust, HHMI, the NIH, or NIGMS.

## Additional information

### Funding

| Funder | Grant reference number | Author |
| --- | --- | --- |
| National Institute of General Medical Sciences | 1R35GM133428-01 | Kelley Harris |
| National Institute of General Medical Sciences | P41GM103533 | Maitreya J Dunham |
| National Institute of General Medical Sciences | R01GM118854 | Alan J Herr |

| Funder | Grant reference number | Author |
|---|---|---|
| Burroughs Wellcome Fund | Career Award at the Scientific Interface | Kelley Harris |
| Kinship Foundation | Searle Scholarship | Kelley Harris |
| Pew Charitable Trusts | Pew Scholarship | Kelley Harris |
| Alfred P. Sloan Foundation | Sloan Fellowship | Kelley Harris |
| National Human Genome Research Institute | T32HG00035 | Anja R Ollodart |
| Howard Hughes Medical Institute | Faculty Scholar Award | Maitreya J Dunham |

The funders had no role in study design, data collection and interpretation, or the decision to submit the work for publication.

### Author contributions

Pengyao Jiang, Conceptualization, Data curation, Formal analysis, Investigation, Methodology, Project administration, Software, Visualization, Writing – original draft, Writing – review and editing; Anja R Ollodart, Methodology, Validation; Vidha Sudhesh, Investigation; Alan J Herr, Methodology, Resources, Supervision, Writing – review and editing; Maitreya J Dunham, Conceptualization, Funding acquisition, Methodology, Resources, Supervision, Writing – review and editing; Kelley Harris, Conceptualization, Funding acquisition, Investigation, Methodology, Project administration, Resources, Software, Visualization, Writing – original draft, Writing – review and editing

### Author ORCIDs

Alan J Herr ⓘ http://orcid.org/0000-0002-9498-0972
Maitreya J Dunham ⓘ http://orcid.org/0000-0001-9944-2666
Kelley Harris ⓘ http://orcid.org/0000-0003-0302-2523

### Decision letter and Author response

Decision letter https://doi.org/10.7554/eLife.68285.sa1
Author response https://doi.org/10.7554/eLife.68285.sa2

## Additional files

### Supplementary files

- Transparent reporting form
- Supplementary file 1. List of strains from the 1011 Genomes used for the mutation spectrum PCA.

### Data availability

Sequencing data have been uploaded to the SRA and approved (Accession numbers PRJNA691686).

The following dataset was generated:

| Author(s) | Year | Dataset title | Dataset URL | Database and Identifier |
|---|---|---|---|---|
| Jiang P, Ollodart A, Sudhesh V, Herr A, Dunham M, Harris K | 2021 | A modified fluctuation assay reveals a natural mutator phenotype that drives mutation spectrum variation within *Saccharomyces cerevisiae* | http://www.ncbi.nlm.nih.gov/bioproject/?term=PRJNA691686 | NCBI BioProject, PRJNA691686 |

The following previously published datasets were used:

| Author(s) | Year | Dataset title | Dataset URL | Database and Identifier |
|---|---|---|---|---|
| Peter J, De Chiara M, Friedrich A, Yue JX, Pflieger D, Bergstrom A, Sigwalt A, Barre B, Freeloads K, Llored A, Cruaud C, Labadie K, Aury JM, Istace B, Lebrigand K, Babry P, Engelen S, Lemainque A, Wincker P, Liti G, Schacherer J | 2018 | Genome evolution across 1,011 Saccharomyces cerevisiae isolates | http://1002genomes. u-strasbg.fr/files/ | Yeast Genomes Website, 1002 |

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
