## [Decision Letter]

[Editors' note: this paper was reviewed by Review Commons.]

**Acceptance summary:**

This article looks at natural variation in mutation rate in yeast, using differences in the pattern of polymorphism between strains to identify likely allelic differences in DNA repair pathways. It is a rare example of population level data and molecular experiments supporting each other, shedding light on a fundamental biological phenomena: mutation and repair.

---

## [Author Response]

We thank the *eLife* editors and reviewers for their many constructive suggestions, which have helped us improve the manuscript substantially. We address questions compiled from the editor as well as updates from our previous response to reviewers below. Moreover, we are pleased to report the results of an extra experiment that we believe considerably strengthens the paper by providing evidence for the mutator activity of the candidate gene *OGG1*.

This manuscripts uses differences in the pattern of polymorphism between various yeast strain to argue that there must have been differences in mutation rates between them, then uses experiments to demonstrate that these differences still exist and have a genetic basis. Understanding mutation rate variation could help us understand the process of mutation, arguably one of the most fundamental processes in genetics and evolution.Everyone was generally happy with your response to the public reviews, with one exception: we strongly suggest you try a reversion-based mutation essay to validate the inferred spectrum. While we find your results convincing, we feel that such a control should at least be attempted, as it would take the paper to a new level in terms of the supporting evidence. We agree that it is possible that the control experiment will not work for your strains, but you would be able to find that out for sure in less than 1 week by simply plating a haploid yeast cultures on 5-FOA plates. If it does not work, so be it, but if it does, the paper will be much stronger.

We appreciate the suggestion regarding performing a reversion-based mutation essay as a validation to our measured mutation spectrum. We attempted to perform such an assay, but we unfortunately encountered many road-blocks along the way that prevented us from doing this successfully in the natural isolates AEQ and AAR. Before detailing these roadblocks, we also wish to point out that for the *URA3* reversion reporter, the *ura3-29* allele can be successfully reverted by C>T, C>A, and C>G mutations (Bulock, Xing, and Shcherbakova 2020), making it complicated to use this reporter to specifically detect whether a strain’s mutation spectrum is enriched for C>A mutations, but a version of *TRP5* has been engineered to specifically detect C>A mutations (Williams et al. 2005).

We tested whether AEQ or AAR might already be auxotrophic for Trp or Ura, but found that they were prototrophic. Therefore, we attempted to use CRISPR to engineer a reversion allele into AEQ and AAR, but we noticed that it failed to replace the locus with the provided template and instead induced a small deletion at the guide RNA cut site. It seems that AEQ and AAR use non-homologous end joining (NHEJ) rather than homologous recombination (HR) to repair double-strand breaks introduced by CRISPR. This phenomenon could be interesting to study further, but also makes it more challenging to perform genomic engineering on these strains. For the above reasons, we are not able to perform the reversion assay as asked. That being said, we are hopeful that our new section verifying the mutagenicity of *OGG1*^AAR/AEQ^ might in part make up for the lack of this confirmatory reversion assay, since it provides additional evidence that AAR and AEQ have a mutator phenotype by showing that the *OGG1* allele present in these strains does not complement the *ogg1* knockout strain as effectively as the laboratory strain allele of *OGG1* is able to do (Line 501-542).

In addition, the Reviewing Editor humbly suggests adding a discussion about similar findings in Arabidopsis "strains", where attempts have also been made to link natural polymorphism patterns with measured mutation rates (including interesting environmental effects). See, e.g.:Weng, Mao-Lun, Claude Becker, Julia Hildebrandt, Manuela Neumann, Matthew T.Rutter, Ruth G. Shaw, Detlef Weigel, and Charles B. Fenster. 2019. "Fine-Grained Analysis of Spontaneous Mutation Spectrum and Frequency in *Arabidopsis thaliana*." Genetics 211 (2): 703-14.Jiang, C., A. Mithani, E. J. Belfield, R. Mott, L. D. Hurst, and N. P. Harberd. 2014. "Environmentally Responsive Genome-Wide Accumulation of de novo *Arabidopsis thaliana* Mutations and Epimutations." Genome Research, October. https://doi.org/10.1101/gr.177659.114.Jiang, Caifu, Aziz Mithani, Xiangchao Gan, Eric J. Belfield, John P. Klingler, Jian-KangZhu, Jiannis Ragoussis, Richard Mott, and Nicholas P. Harberd. 2011. "Regenerant Arabidopsis Lineages Display a Distinct Genome-Wide Spectrum of Mutations Conferring Variant Phenotypes." Current Biology: CB 21 (16): 1385-90.Belfield, Eric J., Carly Brown, Zhong Jie Ding, Lottie Chapman, Mengqian Luo, Eleanor Hinde, Sam W. van Es, et al. 2020. "Thermal Stress Accelerates *Arabidopsis thaliana*Mutation Rate." Genome Research, December. https://doi.org/10.1101/gr.259853.119.

Thanks for these useful and informative references! We have added several of them to the paper’s introduction and Results sections:

Line 142: “Mild environmental stressors, such as high salt, ethanol, and heat, can also alter the mutation rate of *S. cerevisiae* laboratory strains (Liu and Zhang 2019; Voordeckers et al. 2020) as well as *Arabidopsis thaliana* (Jiang et al. 2014; Belfield et al. 2021)”

Line 240: “We note that a discrepancy between de novo and segregating mutation spectra has also been observed in *A. thaliana*, though in the opposite direction, as *A. thaliana* de novo mutations are surprisingly enriched for C>T transitions (Weng et al. 2019).”

Reviewer #1:Summary:The authors describe a useful modified fluctuation assay that couples conventionalmutation rate analysis with mutational spectrum characterization of forward mutations at the *S. cerevisiae* CAN1 locus. They nicely showed that wild yeast isolates display a wide range of mutation rates with strains AAR and AEQ displaying rates ~10-fold higher than the control lab strain. These two strains also showed a bias for C>A mutations, and were the only strains analyzed that had a mutation spectrum statistically different from the lab control. Together, these data provide a compelling proof-of-principle of the applicability of the modified fluctuation analysis approach described in this manuscript. Overall, the manuscript is very well written, and the work reported in it does represent a valuable contribution to the field. However, two primary shortcomings were identified that can be addressed to strengthen the conclusions prior to publication. Both points described below pertain to the analysis of the possible C>A specific mutator phenotype in strains AAR andAEQ.Major comments:1. The work presented in the manuscript does suggest that these two haploids are likely to display the C>A mutator phenotype. Yet, the authors fell short of providing a full and unambiguous demonstration that would elevate the significance of their discovery. They could have directly tested the predicted C>A specific mutator phenotype by conducting additional experiments, one of which is relatively simple. Specifically, they could have performed a simple reversion-based mutation assay to validate the reported C>A mutator phenotype displayed by AAR and AEQ. For example, into AAR, AEQ, and a wild type control, the authors could introduce an engineered auxotrophic marker allele (e.g., ura3 mutation) caused by an A to C substitution, which upon mutation back to A restores prototrophic growth in minimal media (ie. reversion from ura3-C to URA3-A). Such specific reversible allele should be relatively easy to integrate into the AAR and AEQ genomes, as well as in the control strain. Based on the authors' prediction, AAR and AEQ should display a very large increase (far higher than 10 fold) in the reversion rate when compared to a control haploid. To demonstrate the specificity of the mutation spectrum, the authors could test the reversion rates of a different engineered allele requiring a reversion mutation in the opposite direction (ie. reversion from ura3-A to URA3-C). If the AAR and AEQ mutator is specific C>A, one would predict that all three strains should have similar mutation rates for a reversion in the A>C direction. This additional genetic work would thoroughly validate the central discovery and would reinforce the usefulness of the method described in the manuscript.Alternatively, a conventional mutation accumulation and whole genome re-sequencing experiment with parallel lines of AAR, AEQ and a control strain would also very effectively validate the C>A mutator prediction, and it would also answer the authors' discussion point about specificity to the CAN1 locus. However, it would be more costly and much more time consuming.

Please see our response to this issue above.

2. The second concern is in regard to the relatively extensive conclusions drawn about the possible evolutionary significance of the possible C>A mutator in AAR and AEQ. The authors should be more cautious and conservative in the proposed interpretation. As the authors note:'Three of the four C>A-enriched mosaic beer strains, AAR, AEQ, and SACE_YAG, are all haploid derivatives of the [highly heterozygous] diploid *Saccharomyces cerevisiae* var diastaticus strain CBS1782, which was isolated in 1952 from super-attenuated beer.'From this statement, and because the paper cited provided few details on the isolation of CBS1782, it is presumed that these haploid derivatives were most likely isolated as recombinant spores. Furthermore, it is unclear when this isolation occurred, and for how many generations strains AAR and AEQ have been propagated in a haploid state.Herein lies a critical point: AAR and AEQ were recently derived from a diploid background with a "high level of heterozygosity". In a heterozygous diploid context, deleterious point mutations (and any resulting mutator phenotypes) would likely be masked by the presence of wild-type alleles. Now, as haploids, they express a novel genotype (i.e., combination of defective or incompatible parental alleles), which manifests as a mutator phenotype. In this respect, AAR and AEQ appear analogous to the spore derivatives of the incompatible cMLH1-kPMS1 isolate referred to in the manuscript as a notable exception. The analysis of strains harboring incompatible MLH1-PMS1 mutations by Raghavan et al. demonstrated that the heterozygous diploid parents were not themselves mutators, but that haploid spores which had inherited the pair of incompatible alleles displayed mutator phenotype. Collectively, while it can certainly be argued that the strains AAR and AEQ (like the MLH1/PMS1 incompatible strains) are mutators now, this fact alone does not support the conclusion that they have adapted to survive the expression of an extant mutator phenotype. This premise could be tested by analyzing the mutation rates/spectra of four new spores derived from a single tetrad of CBS 1782. Do the four sibling spores display similar or different mutational rates and spectra? If all four spores from a single tetrad exhibit the 10-fold increase in CAN1 mutation rate and the C>A transversion bias, then it can be inferred that the diploid parent is also a mutator in the same manner. Further direct analysis of mutation rates and spectrum in the parent diploid CBS 1782 would complete the work. This finding would be quite significant, and would provide strong evidence that wild strains can in fact tolerate the expression of a chronic mutator allele.

We have now included a new experiment where we sequenced the diploid ancestral strain CBS 1782, called variants and examined polymorphisms together with the 1011 strains (Figure: 5C).

We found that the diploid CBS 1782 also shows the enrichment of C>A mutations that is seen in AEQ and AAR, especially in the rarest category of variants. This result supports the hypothesis that the naturally arising mutator allele is present in the diploid CBS 1782.

To better investigate whether the mutator phenotype is caused at least in part by the *OGG1* allele in AEQ and AAR, we put this allele on a plasmid and used it to transform an *ogg1* knockout strain. The resulting yeast has a higher mutation rate than an *ogg1* knockout transformed with a plasmid containing the reference allele of *OGG1*, suggesting that *OGG1*^AEQ/AAR^ is less effective at DNA repair than *OGG1^wt^*. The *OGG1*^AEQ/AAR^ allele contains two SNPs in AEQ and AAR, both of which are homozygous in CBS 1782, further supporting our hypothesis that the diploid harbors the mutator allele.

We had hoped to measure mutation rates from four haploid derivatives of CBS 1782 and examine their segregation patterns, but this would ideally require sporulation to produce four viable spores. We performed tetrad dissection on CBS 1782 after knocking out its HO locus, but unfortunately found one or two surviving spores out of the four spores produced. As an alternative strategy, we could make the diploid hemizygous for *CAN1* or use a dominant drug resistance marker. We hope to attempt these modifications in a future study, as they require some care in the experimental design and strain engineering.

Minor comments:A final, relatively minor point. That the new haploids AAR and AEQ show distinct mutation rates and spectra opens the door to an interesting line of inquiry, which may help to identify the causative mutator allele in a manner more efficient than searching for missense mutations. It is stated, and it is understandable, that the identification of the possible causal mutations is beyond the scope of the present manuscript. In this spirit, it would be much more appropriate to restrict such considerations to the Discussion section. Specifically, while the authors make a plausible case for OGG1 being a candidate gene responsible for the C>A mutator phenotype, no experimental demonstration was attempted.As such, that text segment should be moved from the Results to the Discussion section.

We have conducted extra experiments to test whether the *OGG1* allele present in AEQ and AAR is responsible for their increase in mutation rate by expressing a plasmid carrying different *OGG1* alleles in an *ogg1* knockout strain (Figure:6). Our results clearly show that the *OGG1*^AEQ/AAR^ allele increases mutation rate (~2.4-fold increase compared to expression of the lab strain allele). The fact that *OGG1* is in the oxidative stress pathway in which null alleles specifically increase C>A mutations adds additional evidence for our hypothesis.

Reviewer #1 (Significance (Required)):As stated in the summary section above, the manuscript by Jiang et al. represents a substantial contribution to the fields of genome stability and genome evolution. The method described is likely to be useful beyond budding yeast. The work will be appreciated by a broad audience of geneticists. The additional work and text modifications proposed above would likely further elevate the impact of this work.Reviewer #2:Mutation is a fundamental force in organismal evolution, and therefore understanding the evolution of mutational mechanisms are important in evolutionary studies. In this manuscript, the authors used strains of *S. cerevisiae* as a model system to study the variations of rates and spectra in mutations with bioinformatic and experimental approaches. First, the authors analyzed the polymorphism data from 1011 strains by PCA analysis and show the variations in spectra. Second, the authors used fluctuation test combined with deep sequencing of the resistance gene to identify mutation rates and spectra in 18 strains, which show ~10-fold mutation rate variations and increased C-to-A mutations in two strains.For the second part, the experimental procedures and statistical analysis are mostly solid. For the first part, as what authors said in the introduction, polymorphism is not equal to the mutation spectra. I think the authors did a good job by being cautious in the wording and having no over-inference after the analysis. It is thus inevitable that the conclusion of this part sounds mostly descriptive. The overall writing is very clear. I will recommend the publication in field-specific journals.Minor comments:P9 – It is very hard to not wonder how the 16 strains were picked in the fluctuation tests. Some comments on that will be appreciated. E.g., was that informed by the results of Figure 1?

We actually did not pick strains based on the results of Figure 1, one reason being that the *CAN1* reporter method only works on haploid strains with a canavanine sensitivity phenotype. We also restricted our analysis to strains without known aneuploidies to maximize our ability to accurately measure the spectra of the strains’ polymorphisms. When possible, given these constraints, we included at least two randomly selected strains from each clade of the 1011 collection whenever possible. These constraints are now explained on page 8:

Line 308: “We used our pipeline to measure mutation rates and spectra in 16 haploid strains from a wide variety of environments (Supplementary File 1C). […] We also selected two lab strains, LCTL1 and LCTL2, to use as controls, since their mutation rates were previously measured”.

P17- In the paragraph "natural selection might contribute.…" , is there any example of "certain mutation types are more often beneficial than others"?

We have added a discussion of this point to the revision (Page: 15).

Line 605: “One example of how natural selection can affect mutation spectrum variation is that transitions are more often synonymous than transversions are (Freeland and Hurst 1998), leading to more frequent selection against transversions in genic regions.”

P20 – Extra '' in the sentence "Adjacent indels were merged if their frequencies differed by less than 10%."

We have fixed this in revision.

In the discussion, it might be good to add a paragraph to compare the rate and spectra reported here and the ones found by MA and then NGS approach(e.g., Zhu et al. 2014).

We have added a reference to the Zhu et al. (2014) spectrum in the discussion (Page 15, Line 633, Figure 4—figure supplement 7), extending our existing comparison of mutation spectra previously reported using *CAN1* (Lang and Murray 2008) and the MA approach (Sharp et al. 2018). Our *CAN1* method also obtains results that are consistent with the Lang et al. 2008 study on the same control strain, with a small (but not significant) discrepancy discussed in the discussion (Line 404, Line 638).

Reviewer #2 (Significance (Required)):The significance of this manuscript will be relatively specific to evolutionary biologists and geneticists, especially those who use yeasts as a model system. For example, I expect the variation of mutation rates and spectra found in this manuscript will impact the following population-genetic analysis in this collection of 1011 strains and motivate more studies on the molecular machineries which affect mutation rates and spectra.In addition, in terms of methodological novelty, adding a novel step of reporter-gene sequencing is a reasonable way to get some information on mutation spectra as it is less labor-intensive than NGS of MAs. Other statistical or experimental procedures in this manuscript mostly follow the approaches which have been developed in previous literature and thus show not much novelty.Reviewer #3:The authors show that certain yeast strains have altered mutation rates/bias. The study is well motivated, genetic variation in mutation rates are not easily uncovered, and capitalizes on yeast and a high-throughput mutation rate/bias method that validates findings of C>A bias from yeast polymorphism data. The results are solid and clearly presented and I have no major concerns.

We appreciate these encouraging comments.

Major comments:

None.

Minor comments:Should have comma: "In addition, environmental.…".

We have fixed this in revision.

Using S. paradoxus to classify derived vs ancestral alleles may not work as well as allele frequency. A 1/100 rare variant is 100x more likely derived than common variant. But with S. paradoxus divergence of say 5%, 5% polymorphic sites are misclassified or NA. Of course, since you used both, this is not a concern. But the number of variants included/excluded in each analysis should be reported. Also, I was a bit surprised that the rare variants are more noisy since most variants are rare.

We agree that the heuristic of classifying rare alleles as derived will do the right thing the majority of the time, but this could potentially create artifactual differences between the mutation spectra of different populations because the exact ratio of rare derived alleles to common derived alleles depends on the population’s demographic history and true site frequency spectrum. If two populations had the same mutation spectrum but very different proportions of variants that are polarized incorrectly, this could create the appearance of a mutation spectrum difference where none exists. In the revision, we report the total number of variants filtered because of the variation present in *S. paradoxus* (Line 187, 189, 191 for all natural variants after filtering, Line 220 for rare variants).

Although the reviewer is correct that most variants are rare, most of the derived variants present in an individual genome are common (since each common variant is present in many genomes). Since each mutation spectrum in the default PCA is computed by counting up the derived variants in an individual genome, these counts are considerably lower and noisier when only rare variants are used or when the common variants are subsampled to randomly assign each one to just a single genome.

In regards to variation in mutation rate based on canavanine resistance. There is a caveat that some strains may be more canavanine resistant – due to differences in transporter abundance or some other aspect of metabolism. Thus, the same mutation would survive and grow (barely) in one strain background, but not another. This caveat is very unlikely to have much of an impact but it would be worth discussing.

Thanks for pointing this out. We also considered the possibility that our mutation rate estimates could be confounded by slight differences in canavanine resistance between strains, and have added results in the revised version (Page 10):

Line 389: “One possible contributor to the observed differences in mutation rates and spectra are differences in target size: the number of distinct *CAN1* mutations that are able to help the strain survive on canavanine media. […] among strains, but it is large enough that we cannot confidently translate our *CAN1*-based mutation rate estimates into genome-wide estimates of the mutation rate per base pair per generation.”

The explanation for synonymous mutations is hitchhikers or errors. However, they could also disrupt translation, here's one possibility PMC4552401.

Thanks for pointing this out. We now mention the possibility of synonymous mutation disrupting translation in Line 366.

Are there CAN allele differences between strains? If there are some, it might be worth mentioning why you do/don't think this influences the mutation rate. E.g. CGG is one step from stop but CGT is not.

We thank the reviewer for pointing to this analysis. We calculated the number of opportunities for missense and nonsense mutations in each strain, and estimated the mutational target size for each strain in the results (Page: 10, Line 389). We find that the overall number of opportunities for missense and nonsense mutations do not differ among strains substantially enough to explain the observed differences between the actual numbers of missense and nonsense mutations.

For the allele counts in Figure 5B. 2 indicates a variant is present in one strain so there are only 9 mutations present in AAR and not found in ANY other strain or just not found in the four listed? Likewise AAR has 36 for count 4, meaning that there are 36 variants present in AAR and one other strain, where other strains are just the 4 shown in the table, or other strains being any of the 1011?

The Figure 5B in the previous version is the Figure 5C in the updated manuscript, with the extra diploid CBS 1782 included as well. We have explained rare allele count more clearly in the revised manuscript (first described in Line 220, which is calculated the same way throughout).

"To our knowledge, this is one of the first" This is an odd way to put it and could be rephrased. As it stand you are either the first and not knowledgeable or knowledgeable and not the first.

We have fixed this in our revision.

"humans, great apes,.." Could you put the citations in the discussion too. I was a little surprised there was no mention of C>A bias as it relates to studies in bacteria and cancer, where there has been a lot of work on mutational spectra. A comment on this literature or whether the C>A biases are not found elsewhere would be nice.

Thanks for this excellent suggestion. We have added a discussion paragraph about some relevant C>A signatures previously identified in cancer as well as the mouse germline:

Line 55: “Many mutator phenotypes in *E. coli* have been linked to defects in DNA repair enzymes (Horst, Wu, and Marinus 1999; Loeb 2001; Prindle, Fox, and Loeb 2010)). Mutator phenotypes also commonly occur in cancer ((Horst, Wu, and Marinus 1999; Loeb 2001; Prindle, Fox, and Loeb 2010), likely either because of relaxed selection against cellular dysfunction or because it is beneficial for cancer cells to adapt rapidly to their aggressive growth niche.”

Line 612: “The C>A mutations enriched in AEQ, AAR, and their relatives might be a signature of oxidative stress damage; such mutations are a known signature of failure to the repair of 8-oxoguanine lesions, which is consistent with a causal role for these strains’ missense substitution in the oxidative stress response gene *OGG1* as which were verified through our *OGG1* plasmid assay. […] SBS18 was initially identified in tumors from individuals with pathogenic variation in *Mutyh* (Viel, et al. 2017), a direct interacting partner of *OGG1* in the 8-oxoguanine repair pathway that has also been implicated in a C>A-dominated germline mutational signature in mice (Sasani, et al. 2021).”

Reviewer #3 (Significance (Required)):I am an evolutionary geneticist with expertise in genomics and bioinformatics. In addition to reviewing papers I also regularly handle papers as an editor. The manuscript provides rare insight into population variation in mutation rates. While differences in mutational biases are well known between species and in some cases within a species, we typically don't know what causes this biases. Environmental factors are often thought to be involved; this work clearly shows that genetic (mutator strains) exist and impact polymorphism in yeast. The manuscript does a nice job in the introduction of explaining the background on mutation rate research and motivation for the work. It also clear explains the advantage of an experimental high-throughput mutation rate/spectra approach. Thus, I believe this new angle on a long-standing problem will be of interest to the community of evolutionary geneticists outside of yeast researchers.

We appreciate the kind comments, which exactly capture the type of impact we hope our study will have.

References:

Belfield, Eric J., Carly Brown, Zhong Jie Ding, Lottie Chapman, Mengqian Luo, Eleanor Hinde, Sam W. van Es, et al. 2021. “Thermal Stress Accelerates *Arabidopsis thaliana* Mutation Rate.” Genome Research 31 (1): 40–50.

Bulock, Chelsea R., Xuanxuan Xing, and Polina V. Shcherbakova. 2020. “Mismatch Repair and DNA Polymerase δ Proofreading Prevent Catastrophic Accumulation of Leading Strand Errors in Cells Expressing a Cancer-Associated DNA Polymerase ϵ Variant.” Nucleic Acids Research 48 (16): 9124–34.

Freeland, S. J., and L. D. Hurst. 1998. “The Genetic Code Is One in a Million.” Journal of Molecular Evolution 47 (3): 238–48.

Horst, J. P., T. H. Wu, and M. G. Marinus. 1999. “*Escherichia coli* Mutator Genes.” Trends in Microbiology 7 (1): 29–36.

Jiang, Caifu, Aziz Mithani, Eric J. Belfield, Richard Mott, Laurence D. Hurst, and Nicholas P. Harberd. 2014. “Environmentally Responsive Genome-Wide Accumulation of de novo *Arabidopsis thaliana* Mutations and Epimutations.” Genome Research 24 (11): 1821–29.

Lang, Gregory I., and Andrew W. Murray. 2008. “Estimating the Per-Base-Pair Mutation Rate in the Yeast *Saccharomyces cerevisiae*.” Genetics 178 (1): 67–82.

Liu, Haoxuan, and Jianzhi Zhang. 2019. “Yeast Spontaneous Mutation Rate and Spectrum Vary with Environment.” Current Biology: CB 29 (10): 1584–91.e3.

Loeb, L. A. 2001. “A Mutator Phenotype in Cancer.” Cancer Research 61 (8): 3230–39.

Prindle, Marc J., Edward J. Fox, and Lawrence A. Loeb. 2010. “The Mutator Phenotype in Cancer: Molecular Mechanisms and Targeting Strategies.” Current Drug Targets 11 (10): 1296–1303.

Sharp, Nathaniel P., Linnea Sandell, Christopher G. James, and Sarah P. Otto. 2018. “The Genome-Wide Rate and Spectrum of Spontaneous Mutations Differ between Haploid and Diploid Yeast.” Proceedings of the National Academy of Sciences of the United States of America 115 (22): E5046–55.

Voordeckers, Karin, Camilla Colding, Lavinia Grasso, Benjamin Pardo, Lore Hoes, Jacek Kominek, Kim Gielens, et al. 2020. “Ethanol Exposure Increases Mutation Rate through Error-Prone Polymerases.” Nature Communications 11 (1): 3664.

Weng, Mao-Lun, Claude Becker, Julia Hildebrandt, Manuela Neumann, Matthew T. Rutter, Ruth G. Shaw, Detlef Weigel, and Charles B. Fenster. 2019. “Fine-Grained Analysis of Spontaneous Mutation Spectrum and Frequency in *Arabidopsis thaliana*.” Genetics 211 (2): 703–14.

Williams, Teresa-Marie, Rebecca M. Fabbri, Jason W. Reeves, and Gray F. Crouse. 2005. “A New Reversion Assay for Measuring All Possible Base Pair Substitutions in *Saccharomyces cerevisiae*.” Genetics 170 (3): 1423–26.

Zhu, Yuan O., Mark L. Siegal, David W. Hall, and Dmitri A. Petrov. 2014. “Precise Estimates of Mutation Rate and Spectrum in Yeast.” Proceedings of the National Academy of Sciences of the United States of America 111 (22): E2310–18.